# Oviductal estrogen receptor α signaling prevents protease-mediated embryo death

**Wipawee Winuthayanon[1,2]\*, Miranda L Bernhardt[1], Elizabeth Padilla-Banks[1], Page H Myers[3], Matthew L Edin[4], Fred B Lih[5], Sylvia C Hewitt[1], Kenneth S Korach[1], Carmen J Williams[1]\***

[1]Reproductive and Developmental Biology Laboratory, National Institute of Environmental Health Sciences, National Institutes of Health, Research Triangle Park, United States; [2]School of Molecular Biosciences, College of Veterinary Medicine, Washington State University, Pullman, United States; [3]Comparative Medicine Branch, National Institute of Environmental Health Sciences, National Institutes of Health, Research Triangle Park, United States; [4]Immunity, Inflammation and Disease Laboratory, National Institute of Environmental Health Sciences, National Institutes of Health, Research Triangle Park, United States; [5]Epigenetics and Stem Cell Biology Laboratory, National Institute of Environmental Health Sciences, National Institutes of Health, Research Triangle Park, United States

**\*For correspondence:**
winuthayanonw@vetmed.wsu.edu
(WW); williamsc5@niehs.nih.gov
(CJW)

**Competing interests:** The authors declare that no competing interests exist.

**Abstract** Development of uterine endometrial receptivity for implantation is orchestrated by cyclic steroid hormone-mediated signals. It is unknown if these signals are necessary for oviduct function in supporting fertilization and preimplantation development. Here we show that conditional knockout (cKO) mice lacking estrogen receptor α (ERα) in oviduct and uterine epithelial cells have impaired fertilization due to a dramatic reduction in sperm migration. In addition, all successfully fertilized eggs die before the 2-cell stage due to persistence of secreted innate immune mediators including proteases. Elevated protease activity in cKO oviducts causes premature degradation of the zona pellucida and embryo lysis, and wild-type embryos transferred into cKO oviducts fail to develop normally unless rescued by concomitant transfer of protease inhibitors. Thus, suppression of oviductal protease activity mediated by estrogen-epithelial ERα signaling is required for fertilization and preimplantation embryo development. These findings have implications for human infertility and post-coital contraception.

## Introduction

In eutherian mammals, fertilization and preimplantation embryo development occur in the oviduct (Fallopian tube in humans), a tubular reproductive tract structure comprised of an inner columnar epithelium supported by mesenchymal tissues including stroma, smooth muscle, and an outer serosa. Oviduct tissue complexity, cellular composition, and luminal fluid components differ along the length of the oviduct and change temporally in response to alterations in steroid hormone levels that occur with the estrous/menstrual cycle (*Buhi et al., 2000*). Estrogen levels are highest in the period immediately prior to ovulation and are decreasing when fertilization occurs in the oviductal ampulla. During the several days of preimplantation embryo development, there is a continued decrease in estrogen and an increase in progesterone. In the uterus, cyclic alterations in steroid hormone levels orchestrate the initial proliferation and then cellular differentiation of the endometrium. These changes are critical for establishment of uterine receptivity to the implanting embryo. It is

**eLife digest** In female mammals, eggs made in the ovaries travel to the uterus via tubes called oviducts (or Fallopian tubes). If sperm fertilize these eggs on the way, they complete this journey as early embryos and then implant into the wall of the uterus. As sperm and then newly fertilized embryos travel down these tubes, they encounter fluid inside the oviduct, which is generated by the cells that line the tube.

The hormonal changes that occur with the menstrual cycle alter the complexity and cellular composition of the uterus. When an egg is fertilized, further changes in the levels of the hormones, estrogen and progesterone, ensure the uterus becomes receptive to the embryo. However, it remains unknown whether such hormone-mediated signals also regulate the oviduct to support fertilization and early embryo development.

To investigate this question, Winuthayanon et al. studied female mice that lack an important estrogen receptor in the cells that line their oviducts and uterus. These mice are infertile. This is partly because most sperm become stuck in the uterus and fail to reach the eggs in the oviduct in order to fertilize them. The oviduct also becomes a hostile environment for both eggs and embryos, as reflected in damaged eggs and the complete loss of all new embryos by two days after fertilization. These embryos die, not because their development fails, but because their outer membrane becomes damaged and breaks apart. Winuthayanon et al. showed that this is due to the persistence of enzymes that form part of the immune system inside the oviduct. These enzymes can degrade proteins and damage cell membranes.

The presence of this estrogen receptor on the inner lining of the oviduct thus appears to be crucially important for reproduction (these effects were not seen when it is removed from other cells of the oviduct). The loss of this receptor also reveals the vital role that estrogen plays in suppressing parts of the immune response to ensure the oviduct provides a supportive environment for fertilization and embryo development. These findings could also have future application in the development of new contraceptives and might also shed light on the causes of human infertility.

unknown what role, if any, steroid hormone-mediated signals have in regulating oviductal function to support fertilization and preimplantation embryo development.

We have shown previously using mice with an epithelial cell-selective ablation of estrogen receptor α (ERα) in the female reproductive tract (*Wnt7a*[Cre];*Esr1*[f/f]) that ERα-mediated crosstalk between stromal and epithelial compartments in the uterus is critical for generating a receptive endometrium (*Winuthayanon et al., 2010*). In the oviduct, ERα is also found in both mesenchymal and epithelial compartments (*Yamashita et al., 1989*), but the contributions of ERα in either of these compartments to oviductal function have not been evaluated. Here, we used epithelial cell (*Wnt7a*[Cre];*Esr1*[f/f]) and stromal cell (*Amhr2*[Cre];*Esr1*[f/-]) selective ERα ablation in mouse models to test the hypothesis that estrogen signaling via ERα in the epithelial cells regulates expression of secreted molecules required to create a microenvironment supportive of fertilization and preimplantation embryo development.

## Results

### Mice lacking ERα in epithelial but not mesenchymal cells have impaired fertilization

Female mice lacking ERα only in reproductive tract epithelial cells were generated by crossing our *Esr1*[f/f] mice (*Hewitt et al., 2010*) with *Wnt7a*[cre] mice (*Winuthayanon et al., 2010*) and are referred to as 'cKO' mice. Mice lacking ERα only in reproductive tract mesenchymal cells were generated by crossing *Esr1*[f/-] with *Amhr2*[Cre] mice (*Huang et al., 2012*) and are referred to as 'mesenchymal cKO' mice. In cKO mice, ERα was not detected in any oviduct epithelial cells, whereas ERα expression in the stromal and muscular layers was no different than in wild-type (WT) mice (control littermates) (*Figure 1*, see also *Figure 1—figure supplement 1*). In mesenchymal cKO mice, ERα was effectively deleted from stromal and muscle cells underlying the epithelium throughout the oviduct (*Figure 1*). Minimal

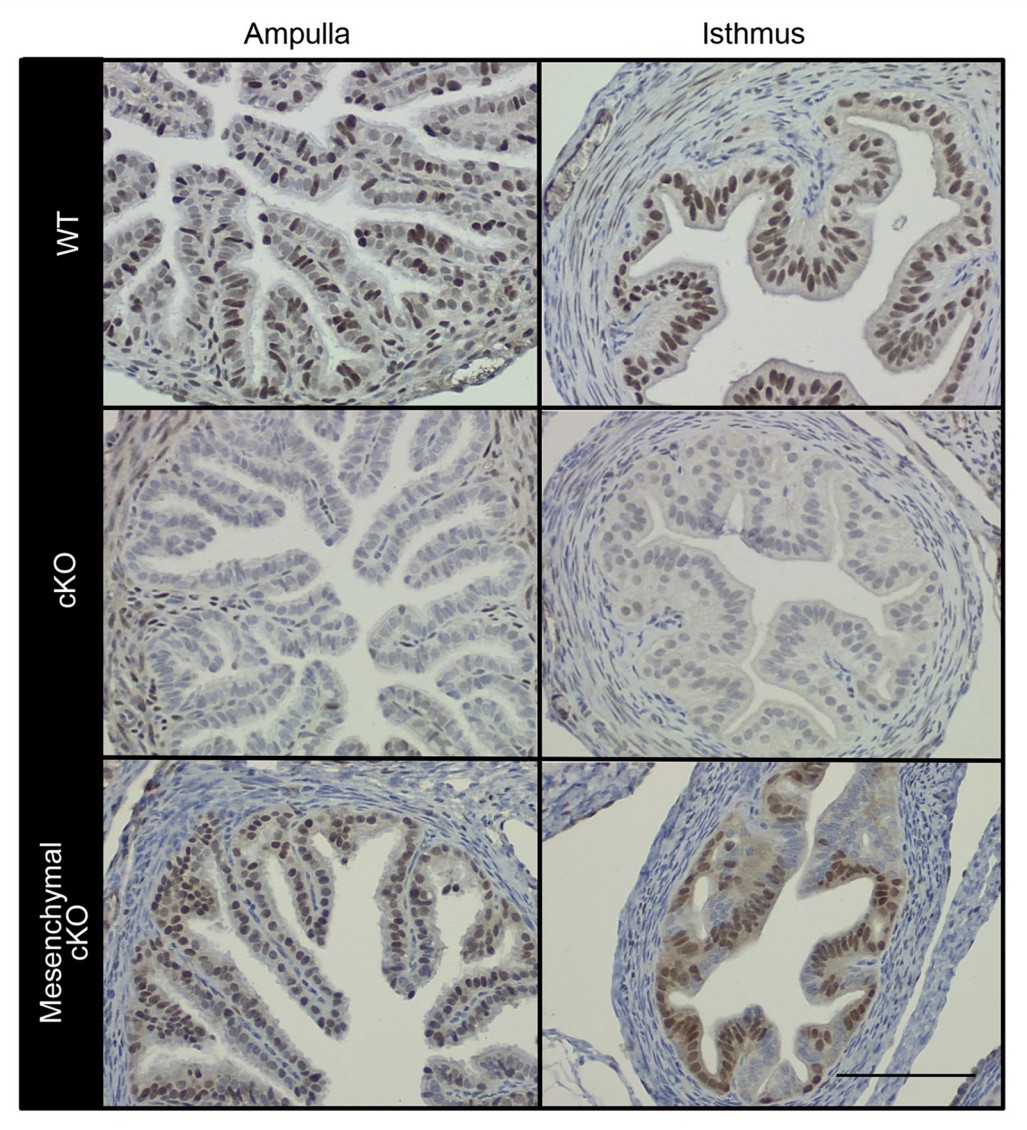

**Figure 1.** Conditional deletion of estrogen receptor α (ERα) from cell type-selective regions of the oviduct. Representative immunohistochemical analysis of ERα in the oviduct regions indicated from wild-type (WT), conditional knockout (cKO), and mesenchymal cKO mice. Scale bar = 100 µm. ERα protein expression in the cKO uterus was reported previously (**Winuthayanon et al., 2010**).

The following figure supplements are available for Figure 1:

**Figure supplement 1.** Additional images showing immunohistochemical analysis of estrogen receptor α (ERα) in oviduct isthmus.

ablation of ERα in the epithelial cells was observed in some regions of the mesenchymal cKO oviduct. No gross or microscopic morphological alterations in oviduct structure were observed in either cKO mouse line. We previously showed that cKO females cycle and mate normally, indicating that gonadotropins and sex steroid hormones are not altered in these mice (**Winuthayanon et al., 2010**). To determine whether lack of ERα in either epithelial or mesenchymal compartments affected fertilization, zygotes (one-cell embryos) were collected from normally cycling, mated females at 0.5 days post coitum (dpc). Despite similar numbers of spontaneously ovulated eggs found in the oviducts (typically 7–8), cKO mice had 58% as many zygotes as WT controls, whereas mesenchymal cKO mice had numbers similar to controls (**Figure 2A,B**). There was also no difference in ovulation efficiency following

**Table 1.** Highly altered genes in cKO compared to WT oviducts at 0.5 dpc.

| Symbol | Entrez gene name | Fold change (cKO vs WT) | p-value |
|---|---|---|---|
| Drd4 | Dopamine receptor D4 | 38.357 | 3.40E-04 |
| Cdh16 | Cadherin 16, KSP-cadherin | 33.796 | 6.04E-08 |
| Clca1 | Chloride channel accessory 1 | 26.786 | 3.40E-03 |
| Lrrc39 | Leucine-rich repeat containing 39 | 24.885 | 1.97E-06 |
| Olfr632 | Olfactory receptor 632 | 23.183 | 1.45E-08 |
| Sct | Secretin | 21.079 | 4.74E-06 |
| Ost alpha | Organic solute transporter alpha | 20.670 | 1.87E-04 |
| Kif12 | Kinesin family member 12 | 19.814 | 1.37E-05 |
| Enthd1 | ENTH domain containing 1 | 18.375 | 2.03E-03 |
| Or8g2 | Olfactory receptor, family 8, subfamily G, member 2 | 17.596 | 5.42E-05 |
| Trp5 | Transient receptor potential cation channel, member 5 | 16.947 | 1.35E-05 |
| Olfr676 | Olfactory receptor 676 | 16.105 | 9.31E-04 |
| Or1s1 | Olfactory receptor, family 1, subfamily S, member 1 | 15.886 | 8.80E-08 |
| Cdh7 | Cadherin 7, type 2 | 15.772 | 6.51E-10 |
| Chrna6 | Cholinergic receptor, nicotinic, alpha 6 | 15.661 | 4.81E-04 |
| Col8a1 | Collagen, type VIII, alpha 1 | 14.885 | 9.07E-05 |
| Olfr470 | Olfactory receptor 470 | 14.542 | 2.39E-03 |
| Prrt3 | Proline-rich transmembrane protein 3 | 14.292 | 8.79E-03 |
| Pla2g5 | Phospholipase A2, group V | 14.256 | 1.84E-07 |
| Slc35f4 | Solute carrier family 35, member F4 | 14.091 | 2.83E-03 |
| Dhrs9 | Dehydrogenase/reductase (SDR family) member 9 | -13.530 | 6.61E-05 |
| Or1j4 | Olfactory receptor, family 1, subfamily J, member 4 | -13.621 | 8.64E-07 |
| Hiat1 | Hippocampus abundant transcript 1 | -13.909 | 9.65E-04 |
| Znf385b | Zinc finger protein 385B | -14.032 | 4.32E-07 |
| Cyp7a1 | Cytochrome P450, family 7, subfamily A, polypeptide 1 | -14.164 | 1.53E-06 |
| Olfr1316 | Olfactory receptor 1316 | -14.225 | 7.13E-07 |
| Cux1 | Cut-like homeobox 1 | -14.331 | 2.87E-04 |
| Expi | Extracellular proteinase inhibitor | -14.373 | 2.52E-06 |
| Sh2d4b | SH2 domain containing 4B | -15.412 | 1.33E-04 |
| Olfr1196 | Olfactory receptor 1196 | -15.449 | 2.26E-05 |
| Thsd7b | Thrombospondin, type I, domain containing 7B | -16.335 | 9.09E-05 |
| Slc7a14 | Solute carrier family 7 (orphan transporter), member 14 | -16.445 | 5.52E-05 |
| Olfr992 | Olfactory receptor 992 | -17.770 | 9.70E-06 |
| Olfr181 | Olfactory receptor 181 | -18.004 | 8.70E-06 |
| Upk1a | Uroplakin 1A | -18.525 | 2.05E-05 |
| Bpifc | BPI fold containing family C | -18.578 | 1.06E-05 |
| C1orf50 | Chromosome 1 open-reading frame 50 | -22.333 | 3.35E-05 |
| Tshr | Thyroid stimulating hormone receptor | -36.758 | 4.18E-06 |
| Dcpp | Demilune cell and parotid protein | -129.642 | 2.22E-03 |
| C6orf15 | Chromosome 6 open-reading frame 15 | -148.254 | 7.15E-05 |

cKO: Conditional knockout; dpc: Days post coitum; WT: Wild-type.

superovulation (*Figure 2C*). Taken together, these findings indicate that lack of ERα in epithelial cells,

but not mesenchymal cells, reduces fertilization.

Impaired fertilization could be explained by effects on sperm, on ovulated eggs, or on both. To determine if lack of epithelial cell ERα affected sperm migration to the site of fertilization, female mice were mated and the numbers of sperm to reach the ovulated cumulus masses in the ampulla and the sperm storage reservoir in the isthmus (*Suarez, 2008*) at 0.5 dpc were counted. Despite high numbers of sperm present in the uterine horns, there was a dramatic reduction in the numbers of sperm to reach both the ampulla and isthmus regions of the oviduct in cKO compared to WT mice (*Figure 2D*, see also *Figure 2—figure supplement 1*), suggesting that effects on sperm migration explained the reduced fertilization. Of note, we found that the mass of sperm in the upper uterine horns of cKO mice was held within a dense, partially solidified mass that retained the tubular shape of the uterine horn following release into culture medium. In contrast, sperm released into culture medium from control uterine horns rapidly dispersed into small clumps. This finding suggested that abnormalities in post-ejaculation seminal fluid processing in the uterine horn caused the failure of sperm migration into the oviduct.

To evaluate whether lack of epithelial ERα could also affect fertilizability of the ovulated eggs, cumulus-oocyte complexes (COCs) were removed from the oviducts and inseminated. In vitro fertilization (IVF) of the eggs from cKO oviducts was still only about 50% as efficient as controls (*Figure 2E*). Removal of the surrounding cumulus cells using hyaluronidase completely rescued IVF of eggs from cKO oviducts to the same level as that of control eggs. Similarly, eggs within intact COCs removed directly from the ovary just prior to ovulation to completely avoid exposure to the cKO oviduct were fertilized in vitro as well as control eggs from WT ovaries. Taken together, these findings indicate that fertilization is impaired in the cKO mouse due to abnormal sperm migration into the oviduct and detrimental effects of the oviduct environment on the cumulus cell masses surrounding the eggs.

## The oviductal microenvironment in mice lacking epithelial ERα is toxic to preimplantation embryos

To evaluate preimplantation embryo development in vivo, oviducts of normally cycling, mated females were flushed at 1.5 dpc and the recovered embryos were counted. Oviducts from WT females contained two-cell embryos (*Figure 2A,B*); however, cKO oviducts had no living embryos. Instead, only remnants of non-viable eggs or embryos were recovered (*Figure 2A*), indicating that all zygotes present at 0.5 dpc died before the two-cell stage. In contrast, mesenchymal cKO oviducts contained similar numbers of embryos with an appearance comparable to controls when collected on both 1.5 dpc (two-cell embryos) and 3.5 dpc (morulae/blastocysts) (*Figure 2A,B*). We conclude that expression of epithelial, but not mesenchymal, ERα in the oviduct is crucial for preimplantation embryo development. Furthermore, when zygotes were recovered from cKO oviducts ~6 hr after fertilization and then cultured in vitro, less than 50% cleaved to the two-cell stage, and only 14% progressed to the blastocyst stage, whereas more than 80% of the zygotes from WT oviducts developed to the blastocyst stage (*Figure 2F*). However, if eggs were collected from either cKO oviducts or ovaries and then fertilized in vitro, the resulting zygotes developed to the blastocyst stage at a rate comparable to WT (*Figure 2G,H*). These results indicate that the zygotes' developmental potential was severely compromised even if they were exposed to the cKO oviduct environment for only a few hours after fertilization.

## Epithelial ERα regulates prostaglandin levels and inflammatory response mediators in the oviduct

To identify factor(s) in the cKO oviduct that could be responsible for inhibiting fertilization and embryo development, we used microarray analysis to compare transcripts in whole oviducts collected at 0.5 and 1.5 dpc from WT and cKO mice, respectively. The most highly altered genes at 0.5 and 1.5 dpc are listed in *Tables 1 and 2*, respectively; representative genes were validated using real-time PCR (*Figure 3*). Using Ingenuity Pathway Analysis software, we found that biological functions significantly altered at 0.5 dpc included tissue development, lipid metabolism, inflammatory response, and cellular growth and proliferation (*Table 3*). At 1.5 dpc, there were fewer altered biological functions than at 0.5 dpc; these functions included tissue development, inflammatory response, cellular movement, and small molecule biochemistry (*Table 4*). Unsupervised hierarchical

**Table 2.** Highly altered genes in cKO compared to WT oviducts at 1.5 dpc.

| Symbol | Entrez gene name | Fold change (cKO vs WT) | p-value |
|---|---|---|---|
| Clca1 | Chloride channel accessory 1 | 34.263 | 2.60E-04 |
| Pcdh8 | Protocadherin 8 | 27.010 | 1.34E-05 |
| Sct | Secretin | 13.070 | 6.05E-06 |
| Myom2 | Myomesin (M-protein) 2, 165kDa | 12.163 | 1.16E-05 |
| Klk8 | Kallikrein-related peptidase 8 | 8.736 | 8.45E-04 |
| Crp | C-reactive protein, pentraxin-related | 7.904 | 4.30E-04 |
| C2orf51 | Chromosome 2 open-reading frame 51 | 6.905 | 3.72E-04 |
| Muc4 | Mucin 4, cell surface associated | 5.932 | 1.02E-04 |
| Cntf | Ciliary neurotrophic factor | 5.643 | 1.02E-05 |
| Csn1s1 | Casein alpha s1 | 5.374 | 2.03E-04 |
| Dbh | Dopamine $\beta$-hydroxylase | 5.248 | 6.50E-07 |
| Hs6st3 | Heparan sulfate 6-O-sulfotransferase 3 | 5.086 | 3.19E-05 |
| G6pc2 | Glucose-6-phosphatase, catalytic, 2 | 4.370 | 6.53E-04 |
| Nr0b1 | Nuclear receptor subfamily 0, group B, member 1 | 4.231 | 2.95E-05 |
| Krt15 | Keratin 15 | 4.115 | 2.60E-04 |
| Kcnd2 | Potassium voltage-gated channel, member 2 | 4.053 | 2.78E-04 |
| Il18r1 | Interleukin 18 receptor 1 | 4.001 | 2.54E-05 |
| Cxcl17 | Chemokine (C-X-C motif) ligand 17 | 3.954 | 2.79E-07 |
| Nrgn | Neurogranin (protein kinase C substrate, RC3) | 3.881 | 2.85E-04 |
| Trvp6 | Transient receptor potential cation channel member 6 | 3.808 | 2.64E-05 |
| Cntnap2 | Contactin-associated protein-like 2 | -4.471 | 1.41E-04 |
| Gpr64 | G-protein-coupled receptor 64 | -4.527 | 2.05E-06 |
| Ly6a | Lymphocyte antigen 6 complex, locus A | -4.585 | 4.28E-04 |
| Unc5cl | Unc-5 homolog C (C. elegans)-like | -4.637 | 6.55E-04 |
| Sbp | Spermine-binding protein | -4.681 | 6.20E-05 |
| Hpgds | Hematopoietic prostaglandin D synthase | -4.790 | 2.61E-05 |
| Galntl5 | UDP-N-acetyl-α-d-galactosamine:polypeptide N-acetylgalactosaminyltransferase-like 5 | -4.804 | 7.22E-04 |
| Gabra5 | Gamma-aminobutyric acid (GABA) A receptor, alpha 5 | -5.309 | 9.81E-05 |
| Sftpd | Surfactant protein D | -5.410 | 1.14E-04 |
| Uox | Urate oxidase, pseudogene | -5.411 | 2.05E-04 |
| Ctse | Cathepsin E | -6.761 | 8.95E-04 |
| Calml3 | Calmodulin-like 3 | -7.842 | 9.63E-07 |
| Sectm1 | Secreted and transmembrane 1 | -8.130 | 3.06E-05 |
| Reg1a | Regenerating islet-derived 1 alpha | -8.255 | 5.03E-05 |
| Upk1a | Uroplakin 1A | -8.839 | 6.84E-05 |
| Rtn1 | Reticulon 1 | -9.943 | 8.23E-06 |
| Mlc1 | Megalencephalic leukoencephalopathy with subcortical cysts 1 | -11.758 | 3.67E-06 |
| Expi | Extracellular proteinase inhibitor | -13.692 | 1.96E-06 |
| Atp6v1c2 | ATPase, H transporting, V1 subunit C2 | -14.225 | 4.67E-06 |
| C6orf15 | Chromosome 6 open reading frame 15 | -253.722 | 3.51E-07 |

cKO: Conditional knockout; dpc: Days post coitum; WT: Wild-type.

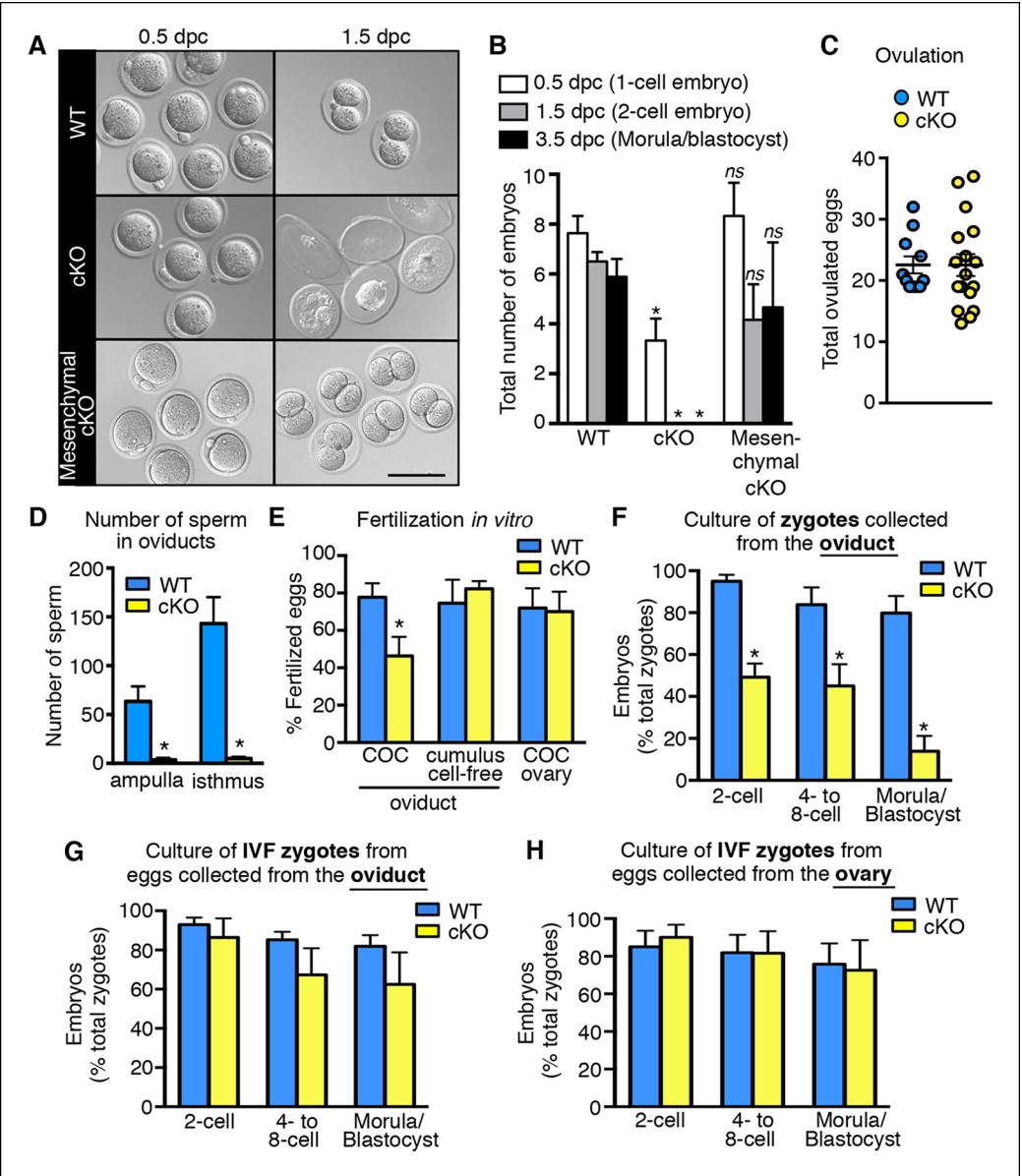

**Figure 2.** Decreased fertilization and increased embryo death in oviducts lacking epithelial estrogen receptor α (ERα). (**A**) Images of zygotes and two-cell embryos collected at 0.5 and 1.5 dpc from each genotype. Scale bar = 100 μm. (**B**) Total embryos collected at the indicated time points (n = 3–11 mice/group). *p< 0.05 vs WT at similar time-point; *ns*, no significant difference vs WT at similar time-point. (**C**) Total ovulated oocytes from WT and cKO females after stimulation with gonadotropins (n = 10–16 mice/group). (**D**) Number of sperm present in the indicated regions of WT and cKO oviducts following mating. Graph shows number of sperm within cumulus cell masses in the ampulla (n = 5 mice/group) and relative number of sperm flushed from the isthmus region (n = 6 mice/group). *, significant difference compared to WT at designated location, Mann–Whitney test, p <0.01. (**E**) IVF efficiency. Cumulus-oocyte complexes (COCs) were collected from the oviducts or ovaries of superovulated WT and cKO females and then inseminated. Cumulus cells were removed from one set of oviduct COCs prior to insemination (cumulus cell-free). Graph indicates the percentage of eggs fertilized out of the total collected (n = 5–7 mice/group). (**F**) Development in vitro of zygotes collected from oviducts of WT and cKO mice. Embryo morphology recorded after 24 hr (two-cell stage), 48 hr (four- to eight-cell stage), and 72 hr (morula and blastocyst stages) (n = 4–5 mice/group). (**G,H**) Development in vitro of zygotes generated by IVF of oocytes from (**G**) oviducts (n = 5–7 mice/group) or (**H**) ovaries of WT and cKO mice (n = 5–7 mice/group). All graphs represent mean ± SEM. *, significant difference compared to WT at designated time point, p<0.05. cKO: Conditional knockout; dpc: Days post coitum; IVF: In vitro fertilization; WT: Wild-type.

*Figure 2. continued on next page*

*Figure 2. Continued*

The following figure supplements are available for Figure 2:

**Figure supplement 1.** Representative images of sperm flushed from oviductal isthmus of wild-type (WT) and conditional knockout (cKO) mice.

clustering of the microarray signal intensities indicated two groups of gene expression patterns, one consisting of WT 0.5 dpc, and the second including all remaining groups (*Figure 4A*). Note that in the cKO samples, replicates from both 0.5 and 1.5 dpc did not all cluster together, unlike WT sample replicates, which clustered together according to the day of pregnancy. Instead, the cKO 0.5 dpc samples were much more similar to both the cKO 1.5 dpc and WT 1.5 dpc samples. These findings demonstrate that estrogen signaling via epithelial ERα has dramatic effects on oviductal gene expression at 0.5 dpc but not 1.5 dpc and are consistent with the normal pattern of estrous cycle estrogen levels that reach a peak prior to ovulation and then drop significantly by 1.5 dpc.

The entire female reproductive tract functions as a mucosal immune barrier and secreted antimicrobial proteins form a significant component of the local innate immune response (*Wira et al., 2011*). Biologic activity of antimicrobial proteins is modulated by secreted proteases and protease inhibitors (PIs), both of which can also serve as antimicrobials in the female reproductive tract and at other mucosal surfaces (*Shaw et al., 2008*; *Wira et al., 2011*; *Aboud et al., 2014*; *Dittmann et al., 2015*). Immune suppression by steroid hormones during mid-cycle creates a window of susceptibility to infection well documented in human tissues (*Wira et al., 2010*). Because the embryos in the cKO oviducts were dying rather than simply failing to develop, we hypothesized that estrogen signaling via epithelial ERα is required to suppress secretion of oviduct antimicrobial molecules that can cause preimplantation embryo death. This idea is consistent with the identification of inflammatory response as a biological category significantly altered in the cKO oviduct at both 0.5 and 1.5 dpc (*Tables 3 and 4*). The altered inflammatory response genes included proteases, PIs, defensins, chemokines (including *Il17*, *Il17rb*, and *Cxcl17*), and enzymes regulating prostaglandin (PG) production such as hematopoietic prostaglandin D synthase (*Hpgds*) (*Figure 4B–D*; *Table 5*). Liquid chromatography-tandem mass spectrometry was used to determine the PG profile of WT and cKO oviducts at

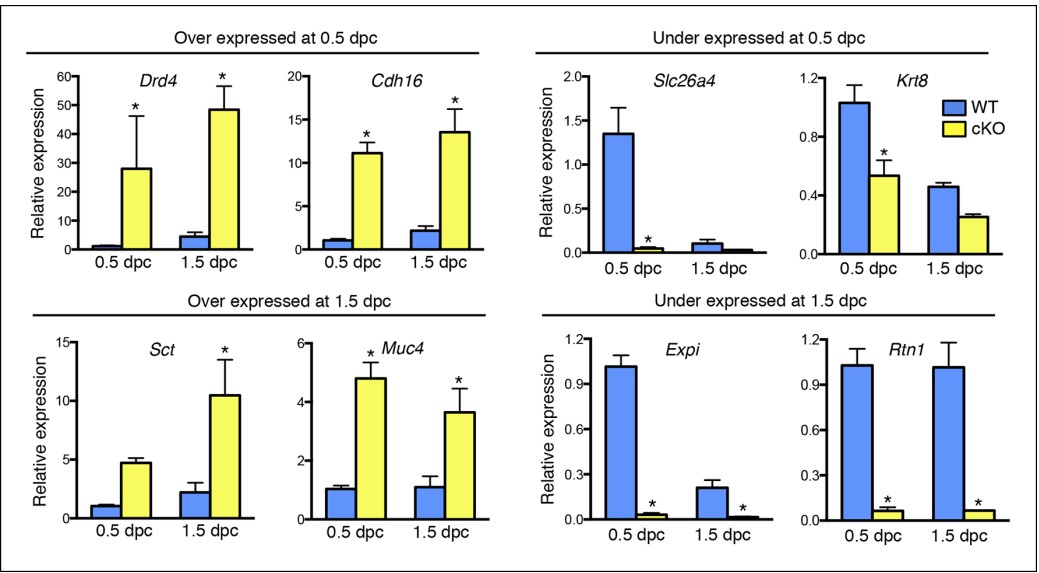

**Figure 3.** Validation of up- and down-regulated genes in conditional knockout (cKO) compared to Wild-type (WT) oviducts at 0.5 and 1.5 dpc using real-time PCR analysis. The transcripts were selected from microarray datasets for over- and under-expression in cKO oviducts compared to WT at 0.5 or 1.5 dpc, as indicated (n = 4–7 mice/group; mean ± SEM). Data represents relative expression level normalized to *Rpl7*. *, significant difference compared to WT at same time point, p<0.05. dpc: Days post coitum.

0.5 dpc. $PGD_2$ and $PGF_{2\alpha}$ levels were significantly lower, whereas $PGE_2$ was increased in cKO compared to WT oviducts (*Figure 4E*). PGs are not normal components of embryo culture medium, so we reasoned that lack of $PGD_2$ and $PGF_{2\alpha}$ could not cause embryo death in vivo. In contrast, $PGE_2$ was elevated in the cKO oviducts and in theory could contribute to the embryo phenotype either directly by effects on the embryo or indirectly by effects on the oviduct. To determine if $PGE_2$ could directly cause embryo death, we added $PGE_2$ to the culture medium during fertilization of WT eggs and during the entire period of embryo development to the blastocyst stage. Inclusion of excess $PGE_2$ had no effect on fertilization or embryo development (*Figure 4F*), suggesting that although epithelial ER$\alpha$ regulates oviduct PG production, these lipid compounds alone are not directly responsible for causing embryo death.

The immune response mediators identified in the microarray analysis were highly enriched for antimicrobial peptides, proteases, and PIs (*Table 5*), all of which have bactericidal activity. At 0.5 dpc, many serine protease transcripts were increased (*Figure 5A*) and serine and cysteine PIs were decreased (*Figure 5B*) in cKO compared to WT oviducts (*Table 5*), suggesting that overall protease activity was elevated. In contrast, expression of proteases and PIs was comparable between WT and

**Table 3.** Selected Ingenuity top biological function categories at 0.5 dpc.

| Category | p-value | # Molecules |
| --- | --- | --- |
| **Tissue development** | | |
| Tissue development | 3.12E-10 | 362 |
| Cell–cell adhesion | 3.33E-04 | 22 |
| Accumulation of monocytes | 8.34E-04 | 5 |
| Angiogenesis of organ | 1.16E-03 | 15 |
| Accumulation of phagocytes | 1.26E-03 | 22 |
| Development of endothelial tissue | 3.61E-03 | 38 |
| Accumulation of eosinophils | 5.73E-03 | 8 |
| **Lipid metabolism** | | |
| Synthesis of lipid | 4.66E-08 | 117 |
| Steroid metabolism | 3.13E-06 | 48 |
| Metabolism of cholesterol | 1.42E-04 | 21 |
| Metabolism of prostaglandin | 1.24E-03 | 27 |
| Synthesis of eicosanoid | 2.02E-03 | 32 |
| Synthesis of prostaglandin D2 | 3.75E-03 | 9 |
| **Inflammatory response** | | |
| Inflammation | 3.52E-04 | 69 |
| Accumulation of monocytes | 8.34E-04 | 5 |
| Accumulation of phagocytes | 1.26E-03 | 22 |
| Accumulation of eosinophils | 5.73E-03 | 8 |
| Accumulation of antigen-presenting cells | 8.80E-03 | 13 |
| **Cellular growth and proliferation** | | |
| Proliferation of endothelial cells | 3.59E-03 | 32 |
| Proliferation of endocrine cells | 4.48E-03 | 14 |
| Proliferation of chondrocytes | 5.56E-03 | 12 |
| Proliferation of epidermal cells | 7.78E-03 | 20 |
| Proliferation of B-lymphocyte-derived cell lines | 8.20E-03 | 19 |
| Proliferation of Th2 cells | 8.70E-03 | 5 |

dpc: Days post coitum.

**Table 4.** Selected Ingenuity top biological function categories at 1.5 dpc.

| Category | p-value | # Molecules |
|---|---|---|
| **Tissue development** | | |
| Tissue development | 3.06E-03 | 44 |
| Development of organ | 1.05E-02 | 30 |
| Aggregation of cells | 1.46E-02 | 8 |
| Organization of tissue | 1.48E-02 | 6 |
| **Inflammatory response** | | |
| Immune response of neutrophils | 4.91E-03 | 4 |
| Chemotaxis of antigen-presenting cells | 6.37E-03 | 5 |
| Immune response of phagocytes | 1.56E-02 | 5 |
| **Cellular movement** | | |
| Mobilization of cells | 6.75E-03 | 4 |
| Mobilization of neutrophils | 8.24E-03 | 2 |
| **Small molecule biochemistry** | | |
| Production of eicosanoid | 3.51E-03 | 7 |
| Synthesis of prostaglandin $E_2$ | 6.13E-03 | 5 |
| Synthesis of lipid | 1.13E-02 | 15 |

dpc: Days post coitum.

mesenchymal cKO oviducts on 0.5 dpc (*Figure 5—figure supplement 1*). These findings suggested that protease activity in the oviduct environment could explain the embryo death phenotype.

## Protease-mediated disruption of plasma membrane integrity leads to embryo death

Treatment with proteases is a well-established method of removing the zona pellucida (ZP), the protective extracellular matrix that surrounds ovulated eggs and developing embryos. Protease treatment results in an initial swelling of the ZP, a morphological change consistent with the ZP appearance of the dead embryos recovered from cKO oviducts (*Figure 2A*). The ZP is comprised of three heavily glycosylated proteins, ZP1, ZP2, and ZP3. After fertilization, there is a physiological proteolytic cleavage of ZP2 when the eggs undergo cortical granule exocytosis and release the protease ovastacin; this change makes the ZP resistant to penetration by additional sperm (*Burkart et al., 2012*). ZP2 cleavage is responsible for the observation of 'zona hardening' after fertilization, defined experimentally by an increased resistance of the ZP to dissolution by proteases in vitro (*Smithberg, 1953*; *Gulyas and Yuan, 1985*).

One of the cysteine protease inhibitors with reduced expression in cKO oviducts was fetuin B (*Table 5*), which has inhibitory activity toward ovastacin (*Dietzel et al., 2013*). We found that fetuin B protein levels were decreased in cKO oviducts (*Figure 5C,D*). In WT oviducts, most fetuin B localized to epithelial cells; the protein was minimally detected in cKO oviducts (*Figure 5E*). Altered expression of fetuin B was not observed in mesenchymal cKO oviducts (*Figure 5—figure supplement 1*), indicating that fetuin B expression is regulated by estrogen-epithelial ERα signaling.

Although the large number of cortical granules released at fertilization causes sufficient ovastacin release to almost completely cleave ZP2 and prevent polyspermy, oocytes also gradually release cortical granules with time following hormone-induced initiation of maturation (*Ducibella et al., 1990*). The consequent release of ovastacin from the cortical granules causes a small but significant amount of ZP2 cleavage and zona hardening when ovulated eggs are cultured in vitro in the absence of serum (which contains fetuin), or in vivo in *Fetub*[-/-] mice (*Schroeder et al., 1990*; *Dietzel et al., 2013*). We took advantage of ZP2 sensitivity to protease-mediated cleavage over time to determine if protease activity within the cKO oviducts was increased. Ovulated eggs were collected 16 hr after human chorionic gonadotropin (hCG) administration, when they would have spent ~4 hr in the

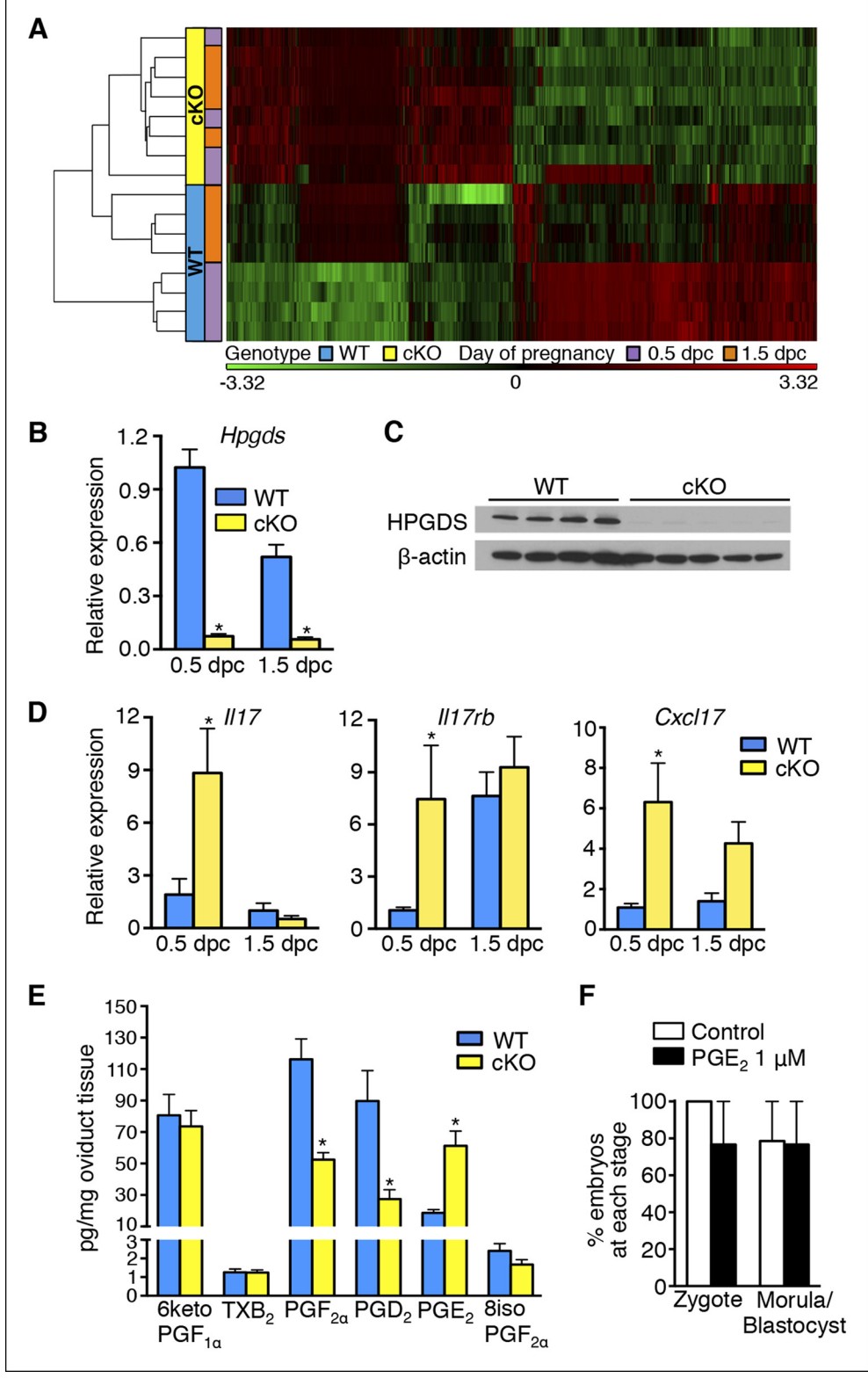

**Figure 4.** Aberrant oviduct innate immune function in the absence of oviductal epithelial estrogen receptor α (ERα). (**A**) Unsupervised hierarchical clustering of microarray data from wild-type (WT) and conditional knockout (cKO) oviducts at 0.5 and 1.5 dpc. Using a 1.5-fold cutoff, 3263 probes were significantly different between WT and cKO oviducts at 0.5 dpc, whereas only 321 probes were different at 1.5 dpc. The heat map shows $log_2$ transformed and standardized g Processed Signals (signal intensities). Green color represents probes

*Figure 4. continued on next page*

*Figure 4. Continued*

with intensity less than mean; red color represents probes with intensity more than mean. Each horizontal bar represents data from a single animal; n = 4 mice/group. (**B**) Real-time PCR of hematopoietic prostaglandin D synthase (*Hpgds*) transcript in WT and cKO oviducts at 0.5 and 1.5 dpc (n = 4–7 mice/group). (**C**) Immunoblot of HPGDS expression in WT and cKO oviducts at 0.5 dpc; β-actin was used as a loading control. Protein lysate from one mouse in each lane; n = 4–5 mice/group. (**D**) Real-time PCR of interleukin-17 (*Il17*), interleukin-17 receptor b (*Il17rb*), and chemokine (CXC motif) ligand 17 (*Cxcl17*)transcripts in WT and cKO oviducts at 0.5 and 1.5 dpc (n = 4–7 mice/group). (**E**) Prostaglandin profile in whole oviduct tissues from WT and cKO at 0.5 dpc. 6ketoPGF$_{1\alpha}$, 6-keto-prostaglandin F$_{1\alpha}$; TXB$_2$, thromboxane B2; PGF$_{2\alpha}$, prostaglandin F$_{2\alpha}$; PGD$_2$, prostaglandin D$_2$; PGE$_2$, prostaglandin E$_2$; and 8isoPGF$_{2\alpha}$, 8-iso-prostaglandin F$_{2\alpha}$ (n = 6–7 mice/group). (**F**) Number of fertilized eggs (zygotes) after insemination in the presence of PGE$_2$ and number of morulae and blastocysts 3 days after treating zygotes with 1 μM PGE$_2$ as compared to vehicle control (n = 36–40 oocytes/group). For all panels, graphs represent mean ± SEM and asterisks indicate significant difference compared to WT at designated time point, $p<0.05$. dpc: Days post coitum; HPGDS: hematopoietic prostaglandin D synthase.

oviduct, and then evaluated for premature ZP2 proteolysis. Based on immunoblot analyses, ovulated eggs recovered from cKO oviducts had significantly less intact ZP2 and more cleaved ZP2 (*Figure 6A–C*). This finding demonstrates that there is a physiologically relevant increase in protease action on the ZP within cKO oviducts in vivo.

To determine whether ovastacin-mediated ZP2 cleavage occurred normally after fertilization, we then examined ZP proteins in zygotes removed from the oviducts ~10 hr after fertilization. As expected, zygotes from WT oviducts had almost complete conversion of ZP2 to cleaved ZP2 (*Figure 6D,E*). In contrast, zygotes from cKO oviducts had much lower ZP2 cleavage, indicating that the normal process of ovastacin-mediated cleavage of the N-terminal region of ZP2 was disrupted. This difference was not due to failure of cortical granule exocytosis because immunofluorescence analysis revealed no differences in cortical granule release or material in the perivitelline space in zygotes from WT and cKO oviducts (*Figure 6F*). Consistent with the lower ZP2 conversion, the ZP of cKO zygotes dissolved twice as quickly as that of WT zygotes when the embryos were subjected to additional protease digestion in vitro using 0.2% α-chymotrypsin (*Figure 6G*). Of note, during the process of ZP dissolution in α-chymotrypsin, zygotes from cKO oviducts underwent swelling and lysis, whereas WT zygotes remained intact for up to an hour after their ZPs were completely dissolved (*Figure 6H* and *Video 1,2*). These findings suggested that the cKO oviduct environment affected not only ZP structure but also zygote plasma membrane integrity.

Increased protease activity in the cKO oviduct could contribute to embryo death by altering the ZP, by acting directly on the zygote plasma membrane, or through a combination of both mechanisms. To test whether proteases are capable of penetrating an intact ZP to access and cleave plasma membrane proteins, a glycosylphosphatidylinositol (GPI)-anchored Enhanced Green Fluorescent Protein (EGFP) that contains a trypsin-sensitive linker sequence (*Rhee et al., 2006*) was expressed in ZP-intact oocytes. The oocytes were imaged while 0.04% α-chymotrypsin was added to the culture medium. EGFP fluorescence at the plasma membrane began to decline within less than 30 s after protease addition and was nearly absent after 2 min (*Video 3*), indicating that α-chymotrypsin rapidly penetrates the ZP to access proteins located at the cell membrane. To determine whether presence of the ZP protects embryos from protease-induced lysis, WT ZP-intact zygotes and WT zygotes with their ZP removed by either mechanical microdissection or by brief exposure to acidic Tyrode's solution were cultured in 0.4% α-chymotrypsin, and time to cell lysis was determined. Regardless of the ZP removal method, ZP-free embryos lysed much more rapidly than ZP-intact embryos (*Figure 6I*). These data show that although α-chymotrypsin is able to rapidly pass through the ZP to access the plasma membrane, the presence of the ZP slows protease-mediated embryo lysis.

The embryo lysis that occurred in vivo in the cKO oviduct and the embryo swelling and lysis that occurred in vitro following protease treatment suggested there were alterations in plasma membrane permeability and ion balance. We first tested whether protease activity alone could disrupt plasma membrane integrity by measuring intracellular sodium ([Na]$_i$) levels, which increase as cells lose the ability to maintain physiological ion gradients (*Bortner et al., 2001*). Zygotes incubated in α-chymotrypsin had increased [Na]$_i$ as compared to vehicle-treated zygotes (*Figure 6J*). Because

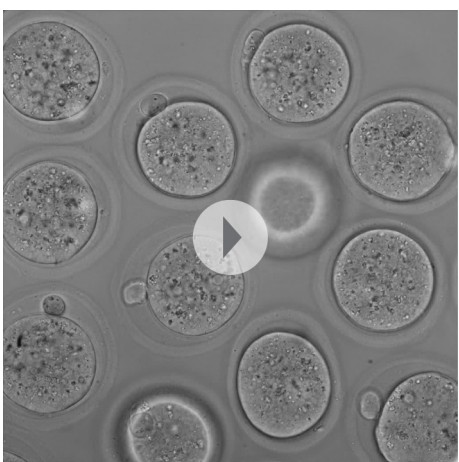

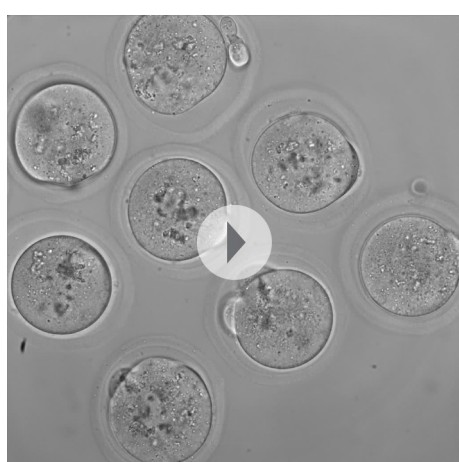

**Video 1.** Morphology of zygotes from WT oviducts during protease treatment in vitro. Zygotes collected from WT oviducts at 24 hr following hCG administration and mating were cultured in PBS containing 0.2% α-chymotrypsin. Images were taken every 5 min for 60 min and are shown at 3 frames/s. hCG: Human chorionic gonadotropin; WT: Wild-type.

**Video 2.** Morphology of zygotes from cKO oviducts during protease treatment in vitro. Zygotes collected from cKO oviducts at 24 hr following hCG administration and mating were cultured in PBS containing 0.2% α-chymotrypsin. Images were taken every 5 min for 60 min and are shown at 3 frames/s. cKO: Conditional knockout; hCG: Human chorionic gonadotropin.

oviducts express numerous antimicrobial peptides, including defensins, that can disrupt permeability of microbial membranes (*Yeaman and Yount, 2003*), we tested whether these molecules could also affect zygote membrane integrity. Zygotes incubated in both α-chymotrypsin and recombinant defensins had further increased $[Na]_i$ when compared to zygotes treated with α-chymotrypsin alone (*Figure 6J*). However, zygotes incubated for several hours in recombinant defensins alone had no morphological evidence of membrane disruption whether or not the ZP was present (*Figure 6—figure supplement 1*). Taken together, these findings indicate that elevated protease activity can disrupt zygote membrane integrity sufficiently to induce embryo lysis, but that antimicrobial molecules may also contribute to the embryo lysis phenotype.

Based on these in vitro studies, we reasoned that if elevated protease activity in the cKO oviducts caused ZP and plasma membrane changes that led to embryo death, then these embryos should be protected by inhibition of protease activity. To test this idea, WT zygotes were transferred into oviducts of WT or cKO recipients at 0.5 dpc using embryo transfer medium that either did or did not contain serine and cysteine PIs. Three days later, the oviducts and uteri were flushed to collect surviving embryos. A significant percentage of zygotes transferred to cKO recipients without PIs were underdeveloped, whereas inclusion of PIs rescued embryo

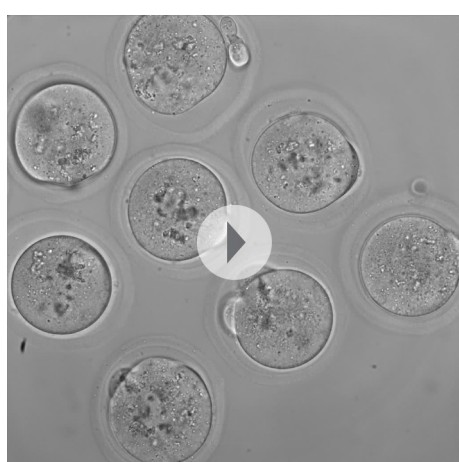

**Video 3.** Protease treatment rapidly cleaves membrane-associated protein despite presence of ZP. Movie shows green fluorescence signal after ZP-intact oocytes expressing a GPI-linked EGFP on the extracellular surface of the plasma membrane were treated with 0.04% α-chymotrypsin. Baseline imaging was performed for 5 min and then imaging was paused for 1 min to allow addition of α-chymotrypsin to the imaging drop. The movie file shows the last frame of baseline imaging, followed by subsequent images taken every 10 s, shown at 3 frames/s. This pattern of fluorescence loss is representative of 5 imaging experiments, each using 6–12 EGFP-GPI-expressing oocytes. (Note that treatment with 0.2% α-chymotrypsin caused complete loss of signal too rapidly to be visualized). ZP: Zona pellucida.

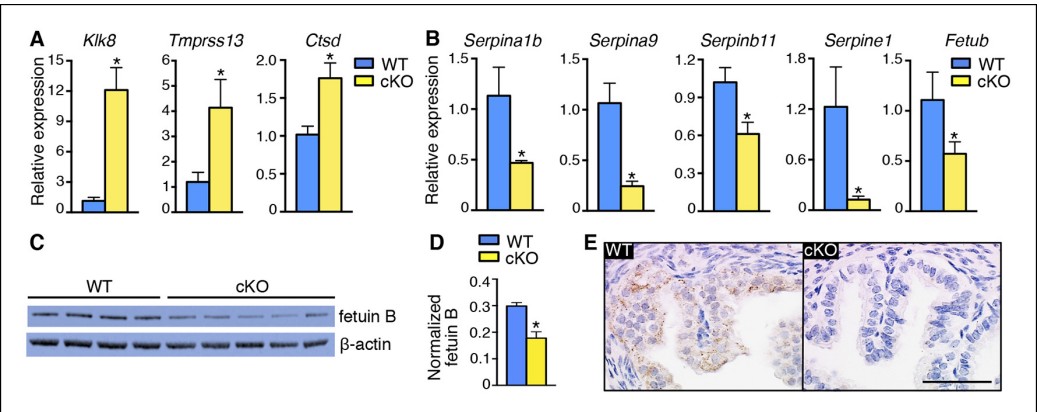

**Figure 5.** Alterations in expression of proteases and protease inhibitors in oviducts lacking epithelial estrogen receptor α (ERα). Real-time PCR of the indicated (**A**) proteases and (**B**) protease inhibitors in wild-type (WT) and conditional knockout (cKO) oviducts at 0.5 dpc (n = 4–7 mice/group; mean ± SEM). (**C**) Immunoblot analysis of fetuin B in WT and cKO oviducts; β-actin served as a loading control. Protein lysate from one mouse in each lane; n = 4–5 mice/group. (**D**) Quantitation of fetuin B signal intensity normalized to β-actin. (**E**) Fetuin B localization in WT and cKO oviducts at 0.5 dpc. Images shown are representative of n = 4 mice/group. Scale bar = 50 μm. For all panels, asterisk indicates significant difference compared to WT, p<0.05. dpc: Days post coitum.

The following figure supplements are available for Figure 5:

**Figure supplement 1.** Comparable expression of proteases and protease inhibitors in wild-type (WT) and mesenchymal conditional knockout (cKO) oviduct at 0.5 dpc.

---

development to control levels (*Figure 7A,B*). These findings indicate that excessive protease activity in the cKO oviduct luminal microenvironment is a proximate cause of the embryo development failure.

## Discussion

In this report, we uncover a pivotal role for epithelial ERα in altering the balance of activities of secreted proteases and PIs in the oviduct to allow successful fertilization and early embryo development. Lack of epithelial ERα in the female reproductive tract results in impaired fertilization by inhibiting sperm migration from the uterus into the oviduct and altering the properties of the cumulus cell mass surrounding the ovulated eggs. The successfully fertilized eggs are exposed to excess oviduct protease activity that disrupts ZP and plasma membrane integrity, causing protease-mediated embryo lysis that may also be promoted by antimicrobial peptides. These findings indicate that the preovulatory increase in estrogen levels not only serves to stimulate ovarian follicle development and uterine endometrial growth, but also has an essential and previously unrecognized function in regulating oviduct physiology.

Global alterations in gene expression critical for implantation have been extensively documented for the uterus (*Reese et al., 2001*; *Kao et al., 2002*). These alterations are programmed by the integrated output of cyclic steroid hormone signals and are mediated by nuclear receptors, mainly in the endometrial stromal and epithelial cells. Our findings that fertilization and preimplantation embryo development are normal in mice lacking mesenchymal ERα indicate that ERα-mediated signaling in this compartment is not a major regulator of oviduct function in early pregnancy. We propose instead that elevated estradiol levels in the periovulatory period act via epithelial ERα to alter secretion of oviduct innate immune mediators, particularly proteases and PIs, and that this process is essential to generate a luminal environment capable of supporting fertilization and embryo development (*Figure 8*).

Several factors likely contribute to the lower fertilization efficiency in the cKO oviducts, but the primary reason appears to be the failure of sperm migration from the uterine horn into the oviduct. Failure of sperm migration may be explained by the abnormal character of the ejaculate present in

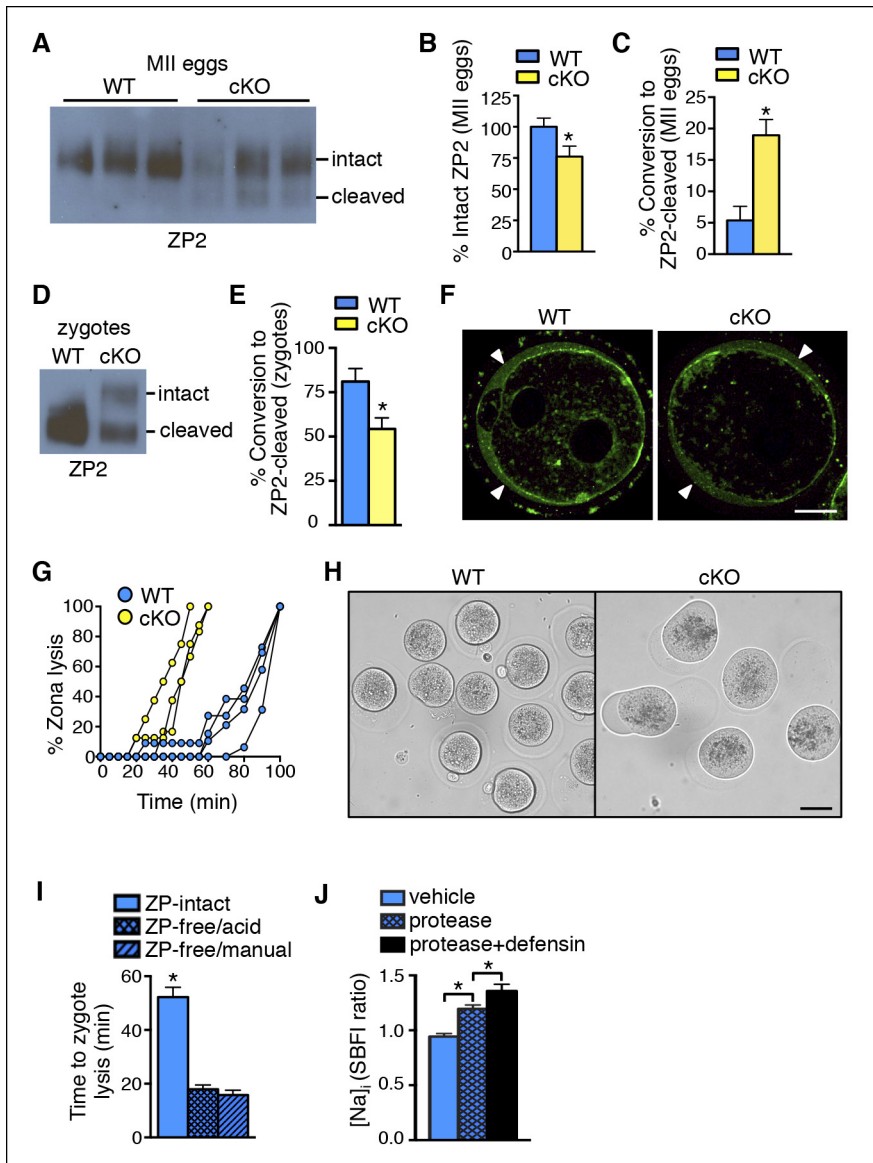

**Figure 6.** Zona pellucida alterations due to elevated protease activity in oviducts lacking epithelial estrogen receptor α (ERα). (**A**) Immunoblot analysis of ZP2 protein in eggs retrieved from wild-type (WT) and conditional knockout (cKO) oviducts ~4 hr after ovulation. Eight eggs from one mouse/lane. (**B,C**) Quantitation of the percentage intact ZP2 protein (**B**) and percentage conversion from intact ZP2 to cleaved ZP2 (**C**) in ovulated eggs from WT and cKO oviducts (n = 6 mice/group); *p<0.05. (**D**) Immunoblot analysis of ZP2 in zygotes retrieved from WT and cKO oviducts ~10 hr after fertilization. Ten zygotes pooled from 3 mice per lane. (**E**) Percentage conversion from intact ZP2 to cleaved ZP2 in zygotes from WT and cKO oviducts. Graph presents data from 7 pools of 10 embryos per group; mean ± SEM. *p <0.05, T-test. (**F**) Images of zygotes from WT and cKO oviducts stained for cortical granules. Arrowheads indicate cortical granule contents in the perivitelline space. Scale bar = 20 μm. (**G**) Percentage ZP lysis over time in zygotes retrieved from WT and cKO oviducts and incubated in 0.2% α-chymotrypsin. Each line represents data from one mouse. (**H**) Images of WT and cKO zygotes after 90 min incubation in 0.2% α-chymotrypsin (n = 3–4 mice/group). Scale bar = 50 μm. (**I**) Time to lysis for zygotes cultured in 0.4% α-chymotrypsin with ZP either intact or removed using treatment with acidic Tyrode's solution or manual microdissection, as indicated. Graph presents data from 15–21 embryos per treatment over three independent experiments; mean ± SEM. *p<0.05, ANOVA. (**J**) $[Na]_i$ in WT zygotes exposed to vehicle, 0.2% α-chymotrypsin (protease), or 0.2% α-chymotrypsin and recombinant defensins (protease defensin). Graph shows relative $[Na]_i$ as indicated by SBFI 340/380 ratio (n = 10–12 embryos/group; mean ± SEM). *p <0.05, ANOVA. cKO: Conditional knockout; MII: Metaphase II; $[Na]_i$: Intracellular sodium; SBFI: Sodium-binding benzofuran isophthalate;WT: Wild-type; ZP: Zona pellucida.

*Figure 6. continued on next page*

*Figure 6. Continued*

The following figure supplements are available for Figure 6:

**Figure supplement 1.** Morphology of WT zygotes after exposure to defensins.

the uterine horns of cKO mice following mating. In rodents, a dense, hard copulatory plug forms in the vagina following mating as a consequence of the protein crosslinking activity of transglutaminase IV, which is generated in the prostate gland and induces coagulation of seminal vesicle proteins (*Dean, 2013*). Although the ejaculate directly enters the rodent uterus, a copulatory plug does not normally form there, so secretions within the uterine lumen must either prevent protein crosslinking or promote liquefaction of the ejaculate. In humans, a copulatory plug does not form; instead, ejaculated semen coagulates into a gelatinous mass that then liquefies. Liquefaction is mediated by the activity of kallikrein 3 (KLK3; better known as prostate-specific antigen), a chymotrypsin-like protease, which promotes release of sperm from the coagulum [reviewed in (*Robert and Gagnon, 1999*)]. Rodents have evolved a large number of kallikrein-related proteins (*Olsson and Lundwall, 2002*), some of which are expressed in the uterus and estrogen-regulated (*Corthorn et al., 1997*; *Rajapakse et al., 2007*). We detected several kallikreins in the oviduct that were highly homologous to human *KLK3*, including *Klk1* and *Klk1b*-family kallikreins, and were downregulated in cKO mice (*Table 5*). Furthermore, we previously demonstrated that estrogen induces *Klk1b5* expression in the uterus of WT but not cKO mice (*Winuthayanon et al., 2014*). These findings suggest that downregulation of KLK3-like proteins in the cKO uterus causes a failure of semen liquefaction, providing a plausible explanation for the failure of sperm migration into the oviduct.

Additional factors within the oviductal environment could also impair fertilization efficiency. Effects of luminal components on the cumulus-oocyte masses could diminish their ability to secrete chemoattractants that promote directional sperm migration (*Eisenbach and Giojalas, 2006*), or the oviductal epithelial cells could secrete alternate chemoattractants that disrupt the directional signal. This idea is consistent with our findings that there were significant alterations in chemokines and their receptors expressed in the cKO oviduct and that there were alterations in PGs, which can modulate chemokine secretion (*Kalinski, 2012*). Effects of altered oviductal PGs, chemokines, proteases, or PIs on the cumulus cells and/or extracellular matrix structure could also contribute to our observation of impaired fertilization in vitro (*Kim et al., 2008*; *Tamba et al., 2008*; *Beek et al., 2012*).

Proteolytic cleavage of ZP2 is a physiological event mediated by ovastacin, a metalloendoprotease contained within the egg's cortical granules that undergo exocytosis at the time of fertilization in the oviduct (*Burkart et al., 2012*). When ZP2 is cleaved, the sperm no longer recognize and bind to the egg's ZP; therefore, ZP2 cleavage promotes monospermic fertilization (*Gahlay et al., 2010*). Ovastacin can also be released prior to fertilization via spontaneous cortical granule exocytosis during meiotic maturation in the ovarian follicle. Studies in the *Fetub*$^{-/-}$ mouse demonstrated that fetuin B, a cysteine PI synthesized in the liver and present in serum, limits ovastacin-mediated premature ZP2 cleavage in the ovarian follicle so that fertilization can occur (*Dietzel et al., 2013*). Here, we have shown that fetuin B is also expressed locally in the oviduct in response to estrogen signaling in the epithelial cells. This finding suggests that in the *Fetub*$^{-/-}$ mouse, complete lack of oviductal fetuin B contributes to the observed fertilization failure. In the ERα cKO mouse, the ~50% reduction in fetuin B explains at least some of the elevation in protease activity in the oviductal lumen.

We found that the cKO oviduct had significant alterations in the expression of a large number of proteases and PIs, in addition to fetuin B. These changes resulted in an overall increase in protease activity within the oviduct milieu documented by the increase in ZP2 cleavage in cKO eggs. Protease action also explains the obvious morphological alterations in ZP structure of the lysed embryos recovered from cKO oviducts because these changes were prevented by artificially reducing protease activity in vivo, a procedure that also rescued embryo development to a large extent. These alterations in overall ZP structure, which were not observed in ovulated eggs recovered after only ~4 hr in the cKO oviducts, could explain the apparent contradiction between increased oviductal protease activity and lower levels of ZP2 cleavage following fertilization in the cKO oviduct. For example, the altered ZP structure could have allowed more rapid passage of ovastacin through the ZP matrix, resulting in lower efficiency of ZP2 cleavage.

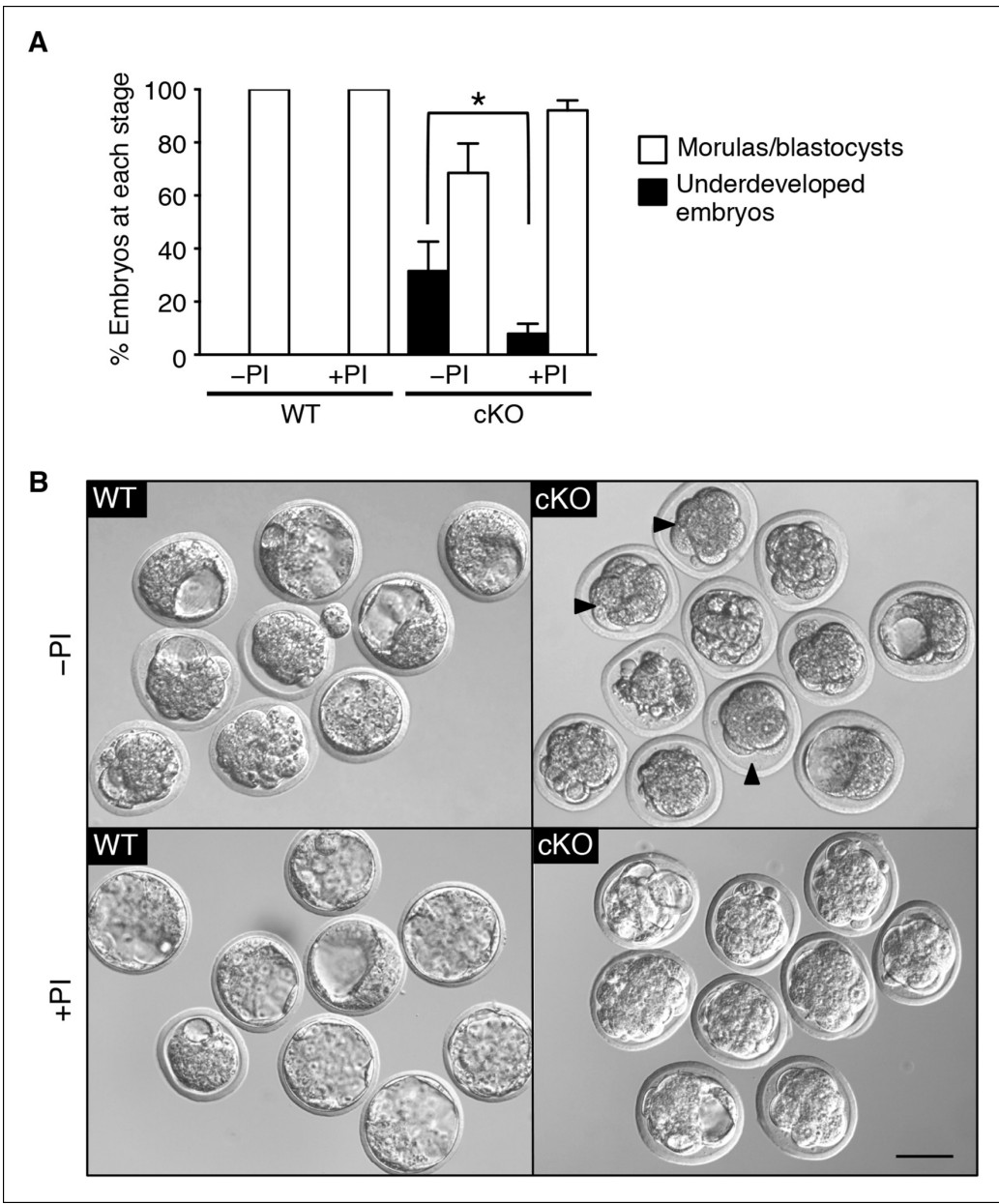

**Figure 7.** Excessive protease activity in vivo in the conditional knockout (cKO) oviduct leads to embryo development failure. (**A**) Percentage of underdeveloped embryos and morula/blastocyst stage embryos retrieved from pseudopregnant wild-type (WT) and cKO recipients that received no protease inhibitors (–PI) or received protease inhibitors ( PI) during embryo transfer (n = 5–12 mice/group and 42–64 embryos/group; mean ± SEM, *p <0.05). (**B**) Representative images of embryos retrieved from WT and cKO recipients at 3.5 dpc in –PI and PI groups. Arrowheads indicate examples of underdeveloped embryos. Scale bars = 50 µm.

ZP thinning or removal is associated with egg and embryo death prior to the two-cell stage in oviducts in vivo but not during culture in vitro (Modlinski, 1970; *Liu et al., 1996*; *Rankin et al., 1996*; *Rankin et al., 2001*). These findings can now be explained by our demonstration that in the presence of a disrupted or absent ZP, embryos lose membrane integrity when exposed to proteases similar to those present in the oviduct during the post-ovulatory time period. Because proteases rapidly penetrate the ZP to access proteins at the plasma membrane, our data suggest that the intact ZP serves as a 'decoy substrate' by providing a large excess of substrate for oviductal proteases that are then less able to disrupt embryo plasma membrane integrity.

 

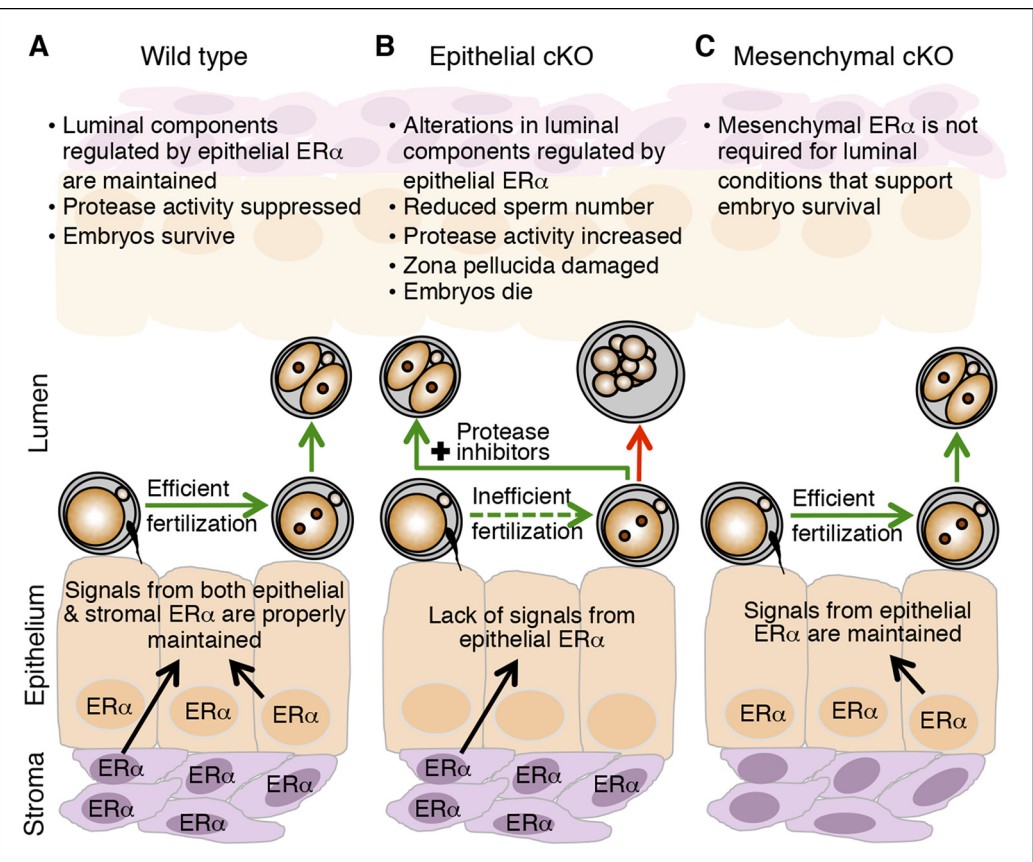

**Figure 8.** Schematic describing how estrogen receptor α (ERα) in oviduct epithelial cells supports fertilization and early embryo development. (A) In wild-type mice, estrogen signals to ERα in both stromal and epithelial cells to suppress secretion of innate immune mediators and generate a luminal environment supportive of sperm migration, fertilization, and preimplantation embryo development. (B) In mice lacking ERα in oviduct epithelial cells, estrogen signaling to stromal cells alone cannot suppress secretion of oviduct immune mediators, resulting in increased protease activity. There is a failure of sperm migration, impaired fertilization, and lysis of successfully fertilized embryos. Embryos can be rescued by inserting protease inhibitors into the oviduct lumen. (C) In mice lacking ERα in oviduct stromal cells, the luminal environment fully supports fertilization and embryo development.

Whether or not antimicrobial peptides such as defensins in the cKO oviduct contribute to the embryo lysis phenotype is less clear. The heavily glycosylated ZP protein matrix has a strong resemblance to bacterial cell walls, which are comprised of a lattice structure of cross-linked peptidoglycans (*Osborn, 1969*). Like the ZP, bacterial cell walls and phospholipid membranes are negatively charged, a property thought to enhance the affinity of antimicrobial peptides that typically have a net positive charge (*Wassarman, 1988*; *Yeaman and Yount, 2003*). These findings indicate that the ZP could serve as a protective barrier for the embryo in part by serving as a 'sink' for antimicrobial peptides. Mammalian phospholipid membranes, in contrast, are generally charge-neutral, which is a property that protects them from antimicrobial peptide actions (*Matsuzaki, 1999*). We showed that the combination of protease and defensins increased zygote membrane permeability to sodium more than protease alone, whereas defensins alone did not affect the zygote plasma membrane. Based on these findings, we speculate that in the cKO oviduct, the initial action of proteases on the zygote plasma membrane alters its charge sufficiently to allow antimicrobial peptides to further disrupt membrane integrity and promote cell lysis.

Infertility in women with hydrosalpinges (inflamed, dilated Fallopian tubes) is partially explained by direct effects of the tubal fluid on embryo development (*Ozmen et al., 2007*). Indeed, culture of mouse zygotes in human hydrosalpingeal fluid leads to zygote degeneration and retarded embryo development (*Mukherjee et al., 1996*; *Beyler et al., 1997*). We found that zygotes fertilized in vivo

**Table 5.** Protease, protease inhibitor, and antimicrobial peptide transcripts in cKO compared to WT oviducts at 0.5 dpc.

| Symbol | Entrez gene name | Fold change (cKO vs WT) | p-value |
|---|---|---|---|
| **Proteases** | | | |
| *Tmprss15* | Transmembrane protease, serine 15 | 16.41 | 2.33E-09 |
| *Klk8* | Kallikrein related-peptidase 8 | 10.34 | 9.85E-04 |
| *Prss42* | Protease, serine, 42 | 9.63 | 7.67E-03 |
| *Prss7* | Protease, serine, 7 (enterokinase) | 8.54 | 2.34E-02 |
| *Klk9* | Kallikrein related-peptidase 9 | 5.62 | 7.06E-06 |
| *Prss33* | Protease, serine, 33 | 5.35 | 5.07E-04 |
| *Prss51* | Protease, serine, 51 | 2.96 | 1.65E-02 |
| *Prss41* | Protease, serine, 41 | 2.69 | 1.36E-04 |
| *Klk7* | Kallikrein related-peptidase 7 | 2.63 | 2.55E-05 |
| *Prss32* | Protease, serine, 32 | 2.45 | 3.10E-02 |
| *Tmprss13* | Transmembrane protease, serine 13 | 2.14 | 6.83E-03 |
| *Cma1* | Chymase 1, mast cell | 2.05 | 2.53E-02 |
| *Prss35* | Protease, serine, 35 | 1.90 | 3.10E-02 |
| *Prss34* | Protease, serine, 34 | 1.89 | 1.92E-02 |
| *Mcpt4* | Mast cell protease 4 | 1.82 | 3.45E-02 |
| *Prss23* | Protease, serine, 23 | 1.66 | 3.54E-02 |
| *Tmprss6* | Transmembrane protease, serine 6 | 1.52 | 4.69E-04 |
| *Ctsd* | Cathepsin D | 1.51 | 4.89E-02 |
| *Prss3* | Protease, serine, 3 | -1.81 | 3.10E-02 |
| *Klk1b3* | Kallikrein 1-related peptidase b3 | -2.08 | 2.86E-02 |
| *Klk1b8* | Kallikrein 1-related peptidase b8 | -2.31 | 2.44E-03 |
| *Prss58* | Protease, serine, 58 | -2.46 | 1.27E-02 |
| *Klk1b26* | Kallikrein 1-related peptidase b26 | -4.01 | 4.57E-02 |
| *Klk1b11* | Kallikrein 1-related peptidase b11 | -4.04 | 1.25E-02 |
| *Klk1* | Kallikrein 1 | -4.08 | 1.11E-03 |
| *Prss29* | Protease, serine, 29 | -4.20 | 3.96E-02 |
| *Klk12* | Kallikrein related-peptidase 12 | -4.68 | 2.05E-03 |
| *Klk1b24* | Kallikrein 1-related peptidase b24 | -6.13 | 3.12E-03 |
| *Klk1b21* | Kallikrein 1-related peptidase b21 | -8.85 | 4.72E-03 |
| *Klk1b27* | Kallikrein 1-related peptidase b27 | -9.65 | 2.72E-03 |
| *Prss28* | Protease, serine, 28 | -28.09 | 2.34E-02 |
| **Protease inhibitors** | | | |
| *Serpini2* | Serine (or cysteine) peptidase inhibitor, clade I (pancpin), member 2 | 6.99 | 3.89E-02 |
| *Serpinb7* | Serine (or cysteine) peptidase inhibitor, clade B (Ovalbumin), member 7 | 6.28 | 2.63E-02 |
| *Serpinb9f* | Serine (or cysteine) peptidase inhibitor, clade B, member 9f | 5.97 | 3.82E-02 |
| *Serpinb12* | Serine (or cysteine) peptidase inhibitor, clade B, member 12 | 2.05 | 1.12E-02 |
| *Serpine2* | Serine (or cysteine) peptidase inhibitor, clade E, member 2 | 1.79 | 4.87E-02 |
| *Cstb* | Cystatin B (stefin B) | -1.76 | 2.25E-04 |

*Table 5. continued on next page*

*Table 5. continued*

| Symbol | Entrez gene name | Fold change (cKO vs WT) | p-value |
|---|---|---|---|
| *Serpinb11* | Serine (or cysteine) peptidase inhibitor, clade B (ovalbumin), member 11 | -2.10 | 2.99E-03 |
| *Fetub* | Fetuin beta | -2.25 | 1.14E-02 |
| *Csta* | Cystatin A (stafin A) | -2.65 | 3.80E-03 |
| *Serpine1* | Serine (or cysteine) peptidase inhibitor, clade E, member 1 | -2.69 | 4.02E-02 |
| *Serpina3b* | Serine (or cysteine) peptidase inhibitor, clade A, member 3B | -3.30 | 2.49E-02 |
| *Serpina1b* | Serine (or cysteine) peptidase inhibitor, clade A, member 1B | -3.49 | 5.80E-03 |
| *Serpina1e* | Serine (or cysteine) peptidase inhibitor, clade A, member 1E | -4.04 | 4.02E-02 |
| *Serpina9* | Serine (or cysteine) peptidase inhibitor, clade A (alpha-1 antiproteinase, antitrypsin), member 9 | -4.22 | 2.52E-03 |
| *Wfdc18* (or *Expi*) | WAP four-disulfide core domain 18 (or extracellular proteinase inhibitor) | -13.69 | 1.96E-06 |
| **Antimicrobial peptides** | | | |
| *Defb8* | Defensin, beta 8 | 13.12 | 2.78E-02 |
| *Defa38* | Defensin, alpha 38 | 3.48 | 2.42E-02 |
| *Defa3* | Defensin, alpha 3 | 2.21 | 3.35E-05 |
| *Defb116* | Defensin, beta 116 | -2.50 | 1.03E-02 |
| *Defb103b* | Defensin, beta 103B | -3.24 | 1.06E-03 |
| *Defa4* | Defensin, alpha 4 | -5.70 | 3.46E-02 |
| *Defb34* | Defensin, beta 34 | -10.02 | 6.15E-05 |

cKO: Conditional knockout; dpc: Days post coitum; WT: Wild-type.

did not survive in the cKO oviduct and that there were detrimental effects on in vitro embryo development only after a several-hour exposure of these zygotes to the cKO oviduct lumen. These findings strongly suggest that innate immune mediators in the luminal fluid of cKO oviducts have persistent effects on embryo development in vitro. The poor but better survival of WT zygotes exposed for 3 days to cKO oviducts in our embryo transfer experiments is likely a result of at least two factors: 1) dilutional effects of the culture medium transferred into the oviduct during the embryo transfer procedure that would partially mitigate effects of proteases and antimicrobial peptides and 2) several hours less time spent in the cKO oviduct because fertilization occurred in a WT oviduct prior to embryo collection and transfer. The critical time frame during which the PIs prevented embryo death was likely to be on the first day of pregnancy, based on the minimal differences in gene expression between cKO and WT oviducts on pregnancy day 1.5 and relatively low circulating estrogen levels on pregnancy days 1.5–3.5.

In summary, we propose that elevated estradiol levels in the preovulatory period act via epithelial ERα to suppress protease-mediated aspects of innate immunity in the oviduct. This response to cyclic steroid hormone levels allows the oviduct to alternate between functioning as a mucosal immune barrier and an environment supportive of fertilization and embryo development. These findings imply that disruption of estrogen signaling, for example by endocrine disruptors or post-coital contraceptives, could prevent pregnancy by interfering with suppression of the oviductal mucosal immune response. Lastly, embryotoxic effects of abnormally elevated innate immune mediators in the Fallopian tubes may contribute to infertility in women with hydrosalpinges, endometriosis, or unexplained infertility.

## Materials and methods

### Animals

CF-1 female mice (6-week old; Harlan Laboratories) and B6D2F1/J male mice (8–12 week old; Jackson Laboratory) were obtained commercially. The generation and genotyping of female reproductive tract epithelial ERα knockout (cKO) mouse model was previously described (*Winuthayanon et al., 2010*). Amhr2$^{Cre}$ mice (*Huang et al., 2012*) were crossed with *Esr1*$^{f/-}$ mice (*Hewitt et al., 2010*) to generate mice with a conditional deletion of ERα in the mesenchyme of the female reproductive tract. Females of *Amhr2*$^{Cre}$ ;*Esr1*$^{f/-}$ genotype were designated as mesenchymal cKO. Heterozygous *Esr1*$^{f/-}$ littermates were used as a control group for mesenchymal cKO mice; the heterozygous phenotype was the same as all WT controls used in this study. All animals were maintained and handled according to NIH Animal Care and Use Committee guidelines.

### Oocyte collection and in vitro fertilization

Adult female mice were superovulated with gonadotropins as previously described (*Jefferson et al., 2009*). COCs were collected either from the ovaries 10 hr after hCG injection by puncturing preovulatory follicles with a 30G needle or from the oviductal ampulla 15 hr after hCG injection. IVF was performed as described previously (*Jefferson et al., 2009*) using either intact COCs or cumulus cell-free eggs obtained by hyaluronidase treatment of oviduct COCs. Fertilization was determined by the presence of two pronuclei (one-cell embryos or zygotes) 6–8 hr after insemination. Embryo development was monitored daily.

### Embryo collection and culture

Spontaneously cycling adult females were housed singly with a B6D2F1/J male overnight. Zygotes were collected from the oviduct at 11:00 am on 0.5 dpc. Two-cell embryos were collected from the oviduct at 9:00 am on 1.5 dpc. Unless otherwise specified, embryos from each mouse were cultured separately in microdrops of potassium simplex optimized medium with amino acids (KSOM/AA; Millipore, Billerica, MA) covered with mineral oil and embryo morphological appearance was documented daily. To test the effects of PG E$_2$ (PGE$_2$) on fertilization and embryo development, oocytes were fertilized in Nunc four-well culture plates (Thermo Scientific, Grand Island, NY) in 400 µL human tubal fluid medium (MR-070-D, Millipore) containing 0.1% ethanol (vehicle control) or 1 µM PGE$_2$ (Cayman Chemical, Ann Arbor, MI). Alternatively, pronuclear stage zygotes from superovulated and mated CF-1 females were cultured in 400 µL KSOM/AA containing 0.1% ethanol or 1 µM PGE$_2$. For these experiments, the medium was not covered with mineral oil to avoid loss of the PGE$_2$ from the aqueous phase.

### GPI-EGFP cRNA synthesis and microinjection

DNA containing an acrosin signal sequence, EGFP, and *Thy1* C-terminal GPI-anchoring sequence was amplified by PCR from the pCX::GFP-GPI2 construct, kindly provided by Anna-Katerina Hadjantonakis (*Rhee et al., 2006*), and cloned into the SalI and XbaI sites of the pIVT expression plasmid (*Igarashi et al., 2007*). The ATG sequence in the SphI site of the pIVT multiple cloning region was mutated to GTG using the QuikChange site-directed mutagenesis kit (Agilent, Santa Clara, CA) to avoid premature initiation of translation. Of note, the linker sequence between EGFP and the GPI-anchor contains a predicted trypsin cleavage site. Oocyte collection, culture, and microinjection were performed as previously described (*Bernhardt et al., 2011*). Oocytes were injected with 5–10 pL of cRNA at a pipette concentration of 0.5 µg/µL.

### Protease and defensin treatments and cell imaging

For comparison of ZP lysis times in vitro, zygotes were collected from superovulated and mated cKO or WT females 24 hr after hCG injection (~10 hr after fertilization) and then cultured in 0.2% α-chymotrypsin in PBS under mineral oil at 37°C as previously described (*Gulyas and Yuan, 1985*). ZP digestion was observed under a light microscope at 5–10 min intervals until all zonae were completely lysed. Zona lysis time [t$_{50}$] was calculated as previously described (*Dietzel et al., 2013*). For intracellular sodium ([Na]$_i$) measurement, WT zygotes were either kept ZP-intact or underwent ZP removal using 0.5% pronase. The zygotes were then loaded for 45 min with 20 µM sodium-

**Table 6.** List of the primer sequences used for real-time RT-PCR reactions.

| Symbol | Entrez gene name | Sequences (Forward ([F]) and Reverse [(R)]:5′ → 3′ |
| --- | --- | --- |
| Cdh16 | Cadherin 16 | F: GCATTGCCCAGGTGCACTGGA<br>R: AAGGGTCCTGGAGGCTGGCT |
| Ctsd | Cathepsin D | F: GACAACAATAGGGTCGGCTT<br>R: GCTGGCTTCCTCTACTGGAC |
| Cxcl17 | Chemokine (C-X-C motif) ligand 17 | F: AAGCCACGGGGACCAACACC<br>R: GGCTTGCAGGAACCAATCTTTGC |
| Drd4 | Dopamine receptor 4 | F: TGGACGTCATGCTGTGCACCG<br>R: GGTCACGGCCACGAACCTGTC |
| Fetub | Fetuin B | F: ACGTCTAGCCTTCTGCGATT<br>R: TCCACTGTAAGCCACTCTGC |
| Hpgds | Hemopoietic prostaglandin D synthase | F: GGACTTACAATCCACCAGAGC<br>R: TCCCAGCCAAATCTGTGTTTT |
| Il17 | Interleukin 17 | F: CTGGAGGATAACACTGTGAGAGT<br>R: TGCTGAATGGCGACGGAGTTC |
| Il17rb | Interleukin 17 receptor B | F: TCAGCGCCCATAACATCCCCA<br>R: ACGTGGTTTAGGCAGCCTGGC |
| Klk8 | Kallikrein related-peptidase 8 | F: GTTCCACCCTCTTCCTCAGA<br>R: CTCCCATGAACAGAAGCAGA |
| Krt8 | Keratin 8 | F: TGAAGAAGGATGTGGACTGTGCCT<br>R: ATGCGGGTCTCCTCGTCATACATT |
| Muc4 | Mucin 4 | F: ACCATGTCTTGGGGAACGTC<br>R: ATGCAGGTGAGGTATTCCTGA |
| Rpl7 | Ribosomal protein L7 | F: AGCTGGCCTTTGTCATCAGAA<br>R: GACGAAGGAGCTGCAGAACCT |
| Rtn1 | Reticulon 1 | F: AACGTCGTCGCGGGAACTGT<br>R: AGCTGCCATACCTGTGGATGCAGT |
| Sct | Secretin | F: CCCACGCCGATGCTACTGCT<br>R: TCTTGGGGTCCTGGGAGGTGC |
| Serpina1b | Serine (or cysteine) peptidase inhibitor, clade A, member 1B | F: ATCACCCGGATCTTCAACAA<br>R: CTCATCGATGGTCAGCACAG |
| Serpina9 | Serine (or cysteine) peptidase inhibitor, clade A, member 9 | F: CAGGTGAGACTCCCTTCCTT<br>R: GTGGGAGGACTCTTGGTTGT |
| Serpinb11 | Serine (or cysteine) peptidase inhibitor, clade B, member 11 | F: TCTTCTGAGTGCAGCCAAGT<br>R: AACGCTGAGGGAGTTCTGTT |
| Serpine1 | Serine (or cysteine) peptidase inhibitor, clade E, member 11 | F: ACCGGAATGTGGTCTTCTCT<br>R: TGCCCTTCTCATTGACTTTG |
| Tmprss13 | Transmembrane protease, serine 13 | F: ATAGGTCGCAATGTCCTTCC<br>R:TCTCAAACCACAGTGGGAAC |
| Tshr | Thyroid stimulating hormone receptor | F: CCTGACAGCTATAGACAACGATGCC<br>R: ACGCTGGTGGAAGACACATCTAGCA |
| Wfdc18 (or Expi) | WAP four-disulfide core domain 18 (or extracellular proteinase inhibitor) | F: TTTGTTCTGGTAGCTTTGATTTTCA<br>R: GCGCCAGGTTTTTCTTTGG |

binding benzofuran isophthalate acetoxymethyl ester (SBFI-AM; Life Technologies, Grand Island, NY) in PBS alone, PBS containing 0.2% α-chymotrypsin, or PBS containing 0.2% α-chymotrypsin and a combination of α-defensin 1 and β-defensin 3 (Abcam, Cambridge, MA) each at 50 µg/mL. [Na]$_i$ was detected using an inverted fluorescent microscope with excitation at 340 and 380 nm as described previously for [Ca$^2$]$_i$ measurement (*Miao et al., 2012*).

Germinal vesicle (GV)-stage oocytes from PMSG-treated CF-1 females were injected with cRNA encoding GPI-linked EGFP targeted to the extracellular surface of the plasma membrane. The oocytes were held overnight in medium containing 10 µM milrinone (Sigma, St. Louis, MO) to

prevent maturation and allow protein expression and then placed in PBS in glass-bottom dishes for imaging. GFP fluorescence was recorded every 3–10 s following α-chymotrypsin addition to a final concentration of 0.04%. Imaging was performed as previously described (*Miao et al., 2012*) except that the excitation wavelength was 470 nm. GV oocytes were used rather than metaphase-II arrested eggs or zygotes because adequate targeting of GPI-linked GFP to the plasma membrane was not observed in eggs or embryos microinjected with this cRNA.

For experiments to determine how presence of the ZP affected time to zygote lysis, zygotes were collected from superovulated and mated CF-1 females. Acid-mediated ZP removal was accomplished by brief exposure to acidic Tyrode's solution (pH 1.6) followed by extensive washing in Leibovitz L-15 medium (Life Technologies) containing 1% calf serum (Atlanta Biologicals, Flowery Branch, GA). Manual ZP removal was accomplished by drilling a slit in the zygote ZP approximately 50 μm × 6 μm using a piezo drill and micromanipulator system followed by gentle pipetting using a 70 μm inner diameter capillary. Time to zygote lysis was measured starting after transfer to drops of PBS containing 0.4% α-chymotrypsin. For comparison of ZP-free and ZP-intact embryos, lysis time was determined by performing ratiometric calcium imaging of fura-2 AM-loaded embryos, as previously described (*Miao et al., 2012*). Time of lysis was indicated by a rise in intracellular calcium prior to lysis, followed by a drop in fluorescence when integrity of the plasma membrane was lost. For comparing ZP removal methods, lysis time was determined from bright-field images captured every 5–7 s.

## Embryo transfer and recovery

Embryos were transferred into the oviduct of recipient females according to published procedures (*Nagy et al., 2003*). Briefly, adult WT and cKO females were mated with vasectomized B6D2F1/J males to generate pseudopregnant recipient females, as determined by the presence of a copulatory plug. Pronuclear stage zygotes were flushed at 10:00 am from the oviducts of superovulated and mated CF-1 donor females. Immediately before transfer, the zygotes were placed into KSOM/AA medium that contained no additive or contained two PIs, 100 μM E64 and 50 μM AEBSF (both from Thermo Scientific). E64 is an irreversible cysteine protease inhibitor and AEBSF is an irreversible serine protease inhibitor. Zygotes (8–12 zygotes per recipient) were transferred along with less than 1 μL of medium into a single oviduct of a pseudopregnant recipient at 0.5 dpc. The zygotes were transferred in medium containing protease inhibitors into 5 WT and 5 cKO pseudopregnant recipients, and in medium alone into 16 WT and 16 cKO pseudopregnant recipients. Three days later (3.5 dpc), the recipients were euthanized and the embryos were flushed separately from both the uteri and oviducts to ensure that all embryos were recovered. Only recipients that had viable embryos, which documented a successful embryo transfer procedure, were included in the results; 4 WT and 6 cKO recipients were excluded. Flushed non-viable embryos were excluded because they could not be distinguished accurately from ovulated, unfertilized eggs generated by the pseudopregnant recipients.

## Oviductal sperm counting

Spontaneously cycling WT and cKO females were mated with *Hspa2*-GFP males (*Brown et al., 2014*). The oviducts were collected at 0.5 dpc (10:00 am). The cumulus cell masses were removed from the ampullary region of each oviduct and placed on slides under cover slips. The number of GFP-positive sperm within the cumulus cell masses from both oviducts was counted for each mouse using an epifluorescence microscope; the total number from both cumulus masses was considered the number of ampullary sperm for one mouse. Sperm entry into the oviductal reservoir was tested using superovulated and mated WT and cKO females. The reproductive tracts were collected from the females with copulatory plugs at 0.5 dpc (15–17 hr after hCG administration). The oviducts were dissected from the uteri as close to the uterine horn as possible, and then flushed from the ampulla toward the isthmus, beginning distal to the cumulus mass, using ~30 μL water via a 30 G needle. The number of sperm flushed from each oviduct was counted in one imaging field using a 40X objective; the two counts were added together to obtain a relative count of isthmic sperm for one mouse. The contents of the uterine horns were also flushed out and examined under a dissecting microscope for the presence of sperm and the character of the ejaculate.

## Microarray analysis and real time RT-PCR

Spontaneously cycling WT and cKO females were mated with B6D2F1/J males. The oviducts were collected at 0.5 and 1.5 dpc and snap frozen in liquid nitrogen. RNA was extracted as previously described (*Hewitt and Korach, 2011*). Gene expression analysis was conducted using Agilent Whole Mouse Genome 4 × 44 multiplex format oligo arrays (Agilent Technologies). The data were deposited in NCBI's Gene Expression Omnibus (Accession #GSE37471). The data were log (2) transformed and normalized using Partek Genomics Suite (Partek Inc., St. Louis, MO). ANOVA was used to detect differentially expressed genes between groups. Gene lists were generated using a false discovery rate < 0.05 and absolute value fold change ≥1.5. Hierarchical clustering was done using Partek's default clustering method. The microarray results were validated by real-time RT-PCR as previously described (*Winuthayanon et al., 2010*). Expression values were calculated as fold change normalized to ribosomal protein L7 (*Rpl7*) expression and relative to WT 0.5 dpc oviduct. The primer sequences are listed in *Table 6*.

## Immunoblot and immunohistochemistry analyses

Spontaneously cycling WT and cKO females were mated with B6D2F1/J males, and oviducts were collected at 0.5 dpc for protein analysis. For oviduct immunoblots, protein was extracted and analyzed as described previously (*Hewitt et al., 2003*). The β-actin (#SC1616-R), fetuin B (#PA5-29468), and HPGDS (#MBS601438) antibodies were purchased from Santa Cruz Biotechnology (Dallas, TX), Thermo Scientific, and MyBiosource (San Diego, CA), respectively, and used at a 1:1000 dilution. Immunohistochemical analyses of oviducts were performed as described previously for uterine tissue (*Winuthayanon et al., 2010*). For immunohistochemical analysis, fetuin B and HPGDS antibodies were diluted at 1:200 and 1:1000, respectively. For ZP immunoblots, MII eggs from COCs were collected 16 hr after hCG injection, freed of cumulus cells using hyaluronidase, washed, and then snap frozen in 5 μL of Tissue Protein Extraction Reagent (Thermo Scientific); zygotes were collected at 5:00 pm on the day of the copulatory plug. An equal volume of reducing sample buffer was added, then the samples were denatured, separated on a 4–12% tris-glycine gel (Novex, Grand Island, NY) and transferred to a polyvinyl difluoride membrane (Life Technologies). Blots were blocked in tris-buffered saline and Tween 20 (TBST) containing 3% bovine serum albumin (BSA) and then incubated overnight at 4°C in primary antibody diluted in blocking solution. Primary ZP antibodies (graciously provided by Dr. Jurrien Dean, NIDDK) were as follows: ZP1, mAB M-1.4 diluted 1:500 (*Rankin et al., 1999*); ZP2, mAB M2c.2 diluted 1:5000 (*Rankin et al., 2003*); ZP3, mAB IE-10 diluted 1:10,000 (*East et al., 1985*). The secondary antibody was horseradish peroxidase-conjugated goat anti-rat IgG (Santa Cruz) diluted 1:10,000 in TBST. Chemiluminescence detection was carried out using SuperSignal West Pico (Thermo Scientific). Percentage of ZP2 protein cleavage was calculated as previously described (*Ducibella et al., 1990*). Immunofluorescence analysis of cortical granules was performed using *lens culinaris* agglutinin as described previously (*Connors et al., 1998*) except that the ZPs were left intact to allow visualization of perivitelline space material.

## Eicosanoid profile analysis

Oviducts were collected at 0.5 dpc from mated WT and cKO females as described above, weighed and immediately frozen in liquid nitrogen. 250 μL of 0.1% acetic acid in 5% methanol was added to each oviduct. Oviducts were homogenized in a Tissuelyzer II (Qiagen, Valencia, CA) for 5 min at 30 Hz. An internal standard including d4-PGE$_2$, 10(11)-epoxyheptadecanoic acid, and 10,11-dihydroxy-nonadecanoic acid (30 ng each) was added to each sample. Lipids were isolated by serial liquid/liquid extractions with ethyl acetate. Ethyl acetate was evaporated under gentle nitrogen flow and samples were reconstituted in 50 μL of 30% ethanol. Oxylipid profile was analyzed by liquid chromatography-tandem mass spectrometry as previously described (*Edin et al., 2011*).

## Statistics

Data were analyzed using GraphPad Prism version 5.0 for Mac OS X. All data are presented as mean ± SEM and evaluated for statistically significant differences (p <0.05) using a two-way ANOVA with Bonferroni's post-hoc test, unless otherwise indicated.

## Acknowledgments

We thank Richard Behringer (University of Texas MD Anderson Cancer Center) for the *Amhr2*-Cre and *Wnt7a*-Cre mice, Mitch Eddy (NIEHS) for the *Hspa2*-GFP mice, Jurrien Dean (NIDDK) for the ZP antibodies, Anna-Katerina Hadjantonakis (Memorial Sloan Kettering Cancer Center) for the GPI-GFP construct, and Yingpei Zhang and Brianna Pockette for technical assistance. This research was supported by the Intramural Research Program of the NIH, National Institutes of Environmental Health Sciences: 1ZIAES025034 (MLE), 1ZIAES050167 (FBL), 1ZIAES70065 (KSK), and 1ZIAES102405 (CJW).

## Additional information

### Funding

| Funder | Grant reference number | Author |
|---|---|---|
| National Institute of Environmental Health Sciences | 1ZIAES70065 | Wipawee Winuthayanon<br>Sylvia C Hewitt<br>Kenneth S Korach |
| National Institute of Environmental Health Sciences | 1ZIAES025034 | Matthew L Edin |
| National Institute of Environmental Health Sciences | 1ZIAES050167 | Fred B Lih |
| National Institute of Environmental Health Sciences | 1ZIAES102405 | Miranda L Bernhardt<br>Elizabeth Padilla-Banks<br>Carmen J Williams |

The funder had no role in study design, data collection and interpretation, or the decision to submit the work for publication.

### Author contributions

WW, Performed majority of experiments, Conception and design, Acquisition of data, Analysis and interpretation of data, Drafting or revising the article; MLB, Performed the ZP/embryo lysis experiments, Conception and design, Acquisition of data, Analysis and interpretation of data, Drafting or revising the article; EPB, Performed the ZP immunoblots and oviductal isthmus sperm counts, Acquisition of data, Analysis and interpretation of data; PHM, Performed embryo transfer experiments, Acquisition of data, Analysis and interpretation of data; MLE, Conducted the eicosanoid profile analysis, Acquisition of data, Analysis and interpretation of data, Drafting or revising the article; FBL, Conducted the eicosanoid profile analysis, Acquisition of data, Analysis and interpretation of data; SCH, KSK, Conception and design, Analysis and interpretation of data, Drafting or revising the article; CJW, Conception and design, Acquisition of data, Analysis and interpretation of data, Drafting or revising the article

### Author ORCIDs

Wipawee Winuthayanon, http://orcid.org/0000-0002-5196-8471
Miranda L Bernhardt, http://orcid.org/0000-0001-5424-5685
Sylvia C Hewitt, http://orcid.org/0000-0001-7713-0805
Carmen J Williams, http://orcid.org/0000-0001-6440-7086

### Ethics

Animal experimentation: This study was performed in strict accordance with the recommendations in the Guide for the Care and Use of Laboratory Animals of the National Institutes of Health. All of the animals were handled according to approved institutional animal care and use committee (IACUC) protocol ASP 01-30 of the National Institute of Environmental Health Sciences (NIEHS). The protocol was approved by the NIEHS Animal Care and Use Committee. All surgery was performed under isofluorane anesthesia, and every effort was made to minimize suffering.

# Additional files

## Major datasets

The following datasets were generated:

| Author(s) | Year | Dataset title | Dataset ID and/or URL | Database, license, and accessibility information |
|---|---|---|---|---|
| Winuthayanon W, Hewitt SC, Padilla-Banks E, Orvis GD, Behringer RR, Williams CJ, Korach KS | 2012 | Role of Estrogen Receptor $\alpha$ in Mouse Oviduct Gene Expression | http://www.ncbi.nlm.nih.gov/geo/query/acc.cgi?acc=GSE37471 | Publicly available at the NCBI Gene Expression Omnibus (Accession no: GSE37471). |

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
