## [Decision Letter]

Thank you for submitting your work entitled "Oviductal estrogen receptor α signaling prevents protease-mediated embryo death" for peer review at *eLife*. Your submission has been favorably evaluated by Janet Rossant (Senior editor and Reviewing editor) and two reviewers.

The reviewers have discussed the reviews with one another and the Reviewing editor has drafted this decision to help you prepare a revised submission

Summary:

This study describes a surprising and novel role for ER-alpha in the mouse oviduct. Conditional knockout of the gene in oviductal epithelial cells but not in the mesenchyme results not only in a decrease in sperm transport to the site of fertilization but also in increased protease activity (and antimicrobial peptides) causes zona rupture and preimplantation embryo lethality. Much research on fertilization and preimplantation development has involved in vitro studies, which constitutes a very artificial environment. A major strength of this study is that it focuses on functional analysis in the oviduct where fertilization and early cleavage events actually occur. The figures and videos are of exceptional quality, and, overall, this manuscript is judged to be valuable contribution to our understanding of oviductal function in fertilization and early development.

Essential revisions:

The work was judged to be for the most part complete and many of the conclusions are well justified by the data. Thus the reviewers do not feel that there are major additional experiments that need to be performed. However, they raise a number of questions that they would like to see in a revised manuscript and for which you may decide that some new experimental data might help in the interpretation.

1) The work on sperm transport is exciting, in that it shows reduced sperm arrival at the oviduct. This coincides with reduced fertilization (subsection “Mice Lacking ERα in Epithelial But Not Mesenchymal Cells Have Impaired Fertilization” et seq.). The question that arises is whether unfertilized oocytes were found in the oviduct. Can the failure of fertilization be due to simple fewer male gametes, or are fewer oocytes transported? Do the authors have any idea where the "missing" sperm are in the cKO oviduct?

2) Removal of cumulus cells improved IVF of COCs recovered from cKO oviducts. Do the authors have any idea as to whether sperm penetration of the COC is compromised, whether they undergo a premature acrosome reaction? Such information would be very useful.

3) If there is an assay that would capture total protease activity and you can obtain oviduct fluid, I would encourage the authors to assay the fluid to ascertain whether the protease activity is higher in the cKO than WT. Such data would permit them to make a statement rather than speculate. Note: it is not essential that this experiment be conducted because it may not be feasible.

4) The authors need to provide more background information/rationale regarding why they looked at ZP2 conversion following maturation. They provide this information the Discussion, but a reader not familiar with this system may be unaware of the maturation-associated proteolysis of ZP2. The same holds for providing background information/rationale regarding the effect of protease treatment on ZP dissolution, i.e., zona hardening.

5) In Figure 1, there appears to be no expression of Esr1 in the isthmus of the oviduct of the Esr1 cKO mice, while it seems to be present in the WT animals.

6) The results presented in Figure 2 are intriguing, as it would seem that the cKO oviduct has an effect on the cumulus complexes. This is not further discussed.

7) The authors dismiss the eicosanoids as effectors of embryo demise because elevated PGE2 did not affect embryos in culture. It is more likely that PGE contributes to the phenotype by its effects on the oviduct itself, and this should be mentioned.

8) The ovostacin/fetuin data appear contradictory. There is a reduction in ZP2 cleavage in the cKO, accompanied by an apparent loss of fetuin, the inhibitor of ovostacin. One might expect that decreased fetuin should result in greater ovostacin activity and thus greater cleavage of ZP2. This should probably be discussed.

9) The defensin experiments (subsection “Protease-Mediated Disruption of Plasma Membrane Integrity Leads to Embryo Death” et seq.) are interesting, but are not judged to be extensive or definitive enough, to justify the conclusion that oviductal innate immune suppression mediated by Esr1 signaling is required for fertilization and preimplantation embryo development. The focus of the discussion and conclusions should be on the microarray findings on proteases and the wealth of functional data on proteases presented in the manuscript. Indeed, the defensins are not featured in the excellent summary slide (Figure 8).

10) The results presented in Figure 7 are somewhat surprising, as they indicate the persistence of exogenous protease inhibitors in the oviduct for a fairly long period of time. It is not clear to what the percentage data refer, how could 100% of the embryos be at the morula/blastocyst stage, while 30% ore underdeveloped in the cKO -PI group.

11) The conclusions about effects on the innate immune system seem to somewhat of a stretch and should be modified.

---

## [Author Response]

*The work was judged to be for the most part complete and many of the conclusions are well justified by the data. Thus the reviewers do not feel that there are major additional experiments that need to be performed. However, they raise a number of questions that they would like to see in a revised manuscript and for which you may decide that some new experimental data might help in the interpretation. 1) The work on sperm transport is exciting, in that it shows reduced sperm arrival at the oviduct. This coincides with reduced fertilization (subsection “Mice Lacking ERα in Epithelial But Not Mesenchymal Cells Have Impaired Fertilization” et seq.). The question that arises is whether unfertilized oocytes were found in the oviduct. Can the failure of fertilization be due to simple fewer male gametes, or are fewer oocytes transported? Do the authors have any idea where the "missing" sperm are in the cKO oviduct?*

Yes, unfertilized oocytes were found in the oviduct. When zygotes were collected from the oviducts of spontaneously cycling, mated cKO females at 11:00 am on 0.5 dpc, we routinely found that about half of the 7-8 total ovulated eggs were fertilized and the rest were not. This point has now been clarified in the text of the Results (first paragraph).

To address the question of the “missing” sperm, we performed additional experiments to determine if the sperm manage to enter the “sperm reservoir” in the isthmus of the oviduct after mating and deposition in the uterus. Wild type and cKO females were superovulated and mated. The following morning, females with copulatory plugs were sacrificed and the reproductive tracts removed. The oviducts were dissected from the uteri as close to the uteri as possible, and then flushed from the ampulla toward the isthmus, beginning distal to the cumulus mass, using ~30 µL water via a 30 G needle. The number of sperm flushed from each oviduct was counted. We found that there were far fewer sperm in the oviductal isthmus in the cKO mice, indicating that migration from the uterus to oviduct was severely impaired. These new findings were added to the graph in Figure 2, and a representative image of the sperm flushed from the oviducts is provided as Figure 2—figure supplement 1.

In addition, the upper uterine horns were flushed to release the luminal contents. We found that the mass of sperm in the upper uterine horns of cKO mice was held within a dense, partially solidified mass that retained the tubular shape of the uterine horn following release into culture medium. In contrast, sperm released into culture medium from control uterine horns rapidly dispersed into small clumps. This finding suggests that in response to estrogen signaling, the normal uterine epithelium secretes components that prevent solidification of the ejaculate in the uterine horn, and thus prevents changes similar to those that form the vaginal copulatory plug. The altered seminal fluid properties in the cKO uterine horns likely explain the decreased numbers of sperm in the oviductal sperm reservoir and the subsequent low numbers of sperm to reach the oviductal ampulla. These findings are now mentioned in the Results (second paragraph) and Discussion (third paragraph); the Methods section was also appropriately revised.

*2) Removal of cumulus cells improved IVF of COCs recovered from cKO oviducts. Do the authors have any idea as to whether sperm penetration of the COC is compromised, whether they undergo a premature acrosome reaction? Such information would be very useful.*

As outlined in our response to question #6 below, we did not tease out the mechanisms that could underlie the reduction in fertilization efficiency of the cumulus cell masses recovered from the cKO oviducts, such as alterations in timing of acrosomal exocytosis. We agree that these would be very interesting and useful experiments to perform, but we felt that they were not as critical as determining the mechanisms underlying the 100% embryo death phenotype in vivo and would require a large number of additional experiments. We have added a new paragraph to the Discussion that explores further the effects of the oviductal environment on both sperm migration and fertilization in vitro (fourth paragraph).

*3) If there is an assay that would capture total protease activity and you can obtain oviduct fluid, I would encourage the authors to assay the fluid to ascertain whether the protease activity is higher in the cKO than WT. Such data would permit them to make a statement rather than speculate. Note: it is not essential that this experiment be conducted because it may not be feasible.*

This was an experiment that we were very eager to perform. Unfortunately, despite numerous attempts, we were only able to collect far less than 1 µL fluid from a mouse oviduct, and we were only able to do that if the fluid was collected shortly following ovulation when the oviduct fluid was likely mixed with follicular fluid. Given the difficulty in obtaining fluid, we did not feel that we could reliably collect samples from WT and cKO oviducts that would provide a valid comparison. Our inability to carry out this experiment led to the alternative experiment of examining the cleavage status of the zona pellucida of the ovulated eggs as an indirect indicator of protease activity in the oviduct, which we showed in Figure 5. Application of CRISPR/Cas9 gene targeting methods in larger size animals such as rats, cows or pigs, which have larger oviducts from which oviductal fluid can be reliably obtained, may allow a direct test of protease activity in oviducts lacking ER alpha in epithelial cells.

*4) The authors need to provide more background information/rationale regarding why they looked at ZP2 conversion following maturation. They provide this information the Discussion, but a reader not familiar with this system may be unaware of the maturation-associated proteolysis of ZP2. The same holds for providing background information/rationale regarding the effect of protease treatment on ZP dissolution, i.e., zona hardening.*

We have added more information regarding ZP2 cleavage and zona hardening in the Results section for the benefit of readers less familiar with zona pellucida changes that occur with age and following fertilization (subsection “Protease-Mediated Disruption of Plasma Membrane Integrity Leads to Embryo Death”, first and third paragraphs).

*5) In Figure 1, there appears to be no expression of Esr1 in the isthmus of the oviduct of the Esr1 cKO mice, while it seems to be present in the WT animals.*

There is minimal staining for ERα in the smooth muscle and stromal cells of the isthmus region of the oviduct in both WT and cKO mice, and there was no difference in the intensity of ERα staining in these cell types in the two groups. The vast majority of the ERα staining in this region is in the epithelial cells, where there is an obvious lack of staining in the cKO. Additional oviductal isthmus sections stained for ERα from WT and cKO mice have been added to the manuscript (Figure 1—figure supplement 1) to provide additional examples of the staining patterns in each group for the reader.

*6) The results presented in Figure 2 are intriguing, as it would seem that the cKO oviduct has an effect on the cumulus complexes. This is not further discussed.*

We were not initially anticipating that the cumulus mass being present or absent would impact on the success of in vitro fertilization. Our finding that fertilization efficiency was far lower in vitro unless the cumulus cells were removed indicated that even a short amount of time spent by the cumulus mass in the oviductal environment was detrimental. One explanation for this finding is that the rather sticky cumulus mass brings with it proteases (or other components) from the oviduct that then negatively impact on the function of sperm that were not exposed to the oviductal environment. An alternate explanation is that the oviductal environment directly affects either the hyaluronic acid matrix between the cumulus cells such that the sperm have more difficulty reaching and penetrating the zona pellucida, or causes alterations in the expression of chemoattractant molecules by the cumulus cells. Either explanation is plausible, and they are not exclusive. Because of this reviewer comment, we have now added a new paragraph to the Discussion that explores further the effects of the oviductal environment on both sperm migration and fertilization in vitro (fourth paragraph).

However, teasing out these details was not something we pursued because the finding was related to an in vitro assay, was a partial effect (IVF was about 50% as efficient), and was likely to be multifactorial and so would require extensive experiments. The failure of sperm migration into the oviduct was so dramatic that it seemed much more likely that this was the explanation for the low fertilization efficiency in vivo, rather than an effect of the oviductal environment on the cumulus cell matrix.

*7) The authors dismiss the eicosanoids as effectors of embryo demise because elevated PGE2 did not affect embryos in culture. It is more likely that PGE contributes to the phenotype by its effects on the oviduct itself, and this should be mentioned.*

We were testing whether the eicosanoids could directly cause embryo lysis, which they did not appear to do. But yes, it is certainly possible that the eicosanoids could affect oviductal function in a way that promotes embryo lysis. We have added text to the Results to raise this possibility (subsection “Epithelial ERα Regulates Prostaglandin Levels and Inflammatory Response Mediators in the Oviduct”, second paragraph).

*8) The ovostacin/fetuin data appear contradictory. There is a reduction in ZP2 cleavage in the cKO, accompanied by an apparent loss of fetuin, the inhibitor of ovostacin. One might expect that decreased fetuin should result in greater ovostacin activity and thus greater cleavage of ZP2. This should probably be discussed.*

The apparent contradiction in our findings that fetuin B was decreased and ZP2 cleavage was also decreased is now addressed in the Discussion (sixth paragraph).

*9) The defensin experiments (subsection “Protease-Mediated Disruption of Plasma Membrane Integrity Leads to Embryo Death” et seq.) are interesting, but are not judged to be extensive or definitive enough, to justify the conclusion that oviductal innate immune suppression mediated by Esr1 signaling is required for fertilization and preimplantation embryo development. The focus of the discussion and conclusions should be on the microarray findings on proteases and the wealth of functional data on proteases presented in the manuscript. Indeed, the defensins are not featured in the excellent summary slide (Figure 8).*

We agree that the defensin data is not extensive or definitive, and does not belong in the summary schematic because we have no evidence that it represents a functional difference between the WT and cKO oviducts. The expression levels of various defensins were different in the WT and cKO oviducts, but there was not an overall up- or down-regulation of defensin expression in the cKO oviduct, and even direct application of defensins to embryos in vitro did not cause embryo lysis (Figure 6—figure supplement 1), so it is unlikely that defensins alone explain the embryo lysis phenotype. Instead, we believe that the information provided regarding how defensins could contribute to the protease-mediated embryo lysis phenotype brings out the multifaceted nature of the innate immune environment present in the oviduct lumen.

Our conclusions regarding oviductal innate immune suppression are based on our understanding that both proteases and protease inhibitors function as important components of the innate immune system. For example, kallikreins are proteases that function to activate antimicrobial peptides (reviewed in Yu et al., Biol. Chem. 2014, 395:931) and have been documented to have this function in the human female reproductive tract (Shaw et al., Biol Chem 2008, 389:1513). SLPI and elafin are protease inhibitors that have antibacterial, antifungal, and antiviral activities (see Lecaille et al., Biochimie 2015, doi:10.1016/j.biochi.2015.08.014). Proteases and protease inhibitors both serve as innate immune mediators in the female reproductive tract (Wira et al., J Reprod Immunol 2011, 65:196; Aboud et al., American Journal of Reproductive Immunology 2014, 71:12). One of the downregulated protease inhibitors we identified in cKO oviducts, *Serpine1*, was recently demonstrated to function as a secreted protease inhibitor that inhibits influenza A virus spreading by preventing viral surface glycoprotein maturation (Dittmann et al., Cell 2015, 160:631). In response to this reviewer concern, we have included additional information regarding antimicrobial proteins in the Results section (subsection “Epithelial ERα Regulates Prostaglandin Levels and Inflammatory Response Mediators in the Oviduct”, second paragraph) and modified the Abstract, Significance Statement, and the first and second paragraphs of the Discussion to focus more on the protease/protease inhibitor findings.

*10) The results presented in Figure 7 are somewhat surprising, as they indicate the persistence of exogenous protease inhibitors in the oviduct for a fairly long period of time. It is not clear to what the percentage data refer, how could 100% of the embryos be at the morula/blastocyst stage, while 30% ore underdeveloped in the cKO -PI group.*

We agree it is highly unlikely that the exogenous protease inhibitors would persist in the oviduct for 3 days. However, we found by microarray analysis that in contrast to the large differences in gene expression in WT and cKO oviducts on pregnancy day 0.5, there was very little difference in the gene expression patterns on pregnancy day 1.5. Although we did not perform microarrays on pregnancy days 2.5 or 3.5, we would predict that on these days the oviduct would have similarly small differences because estradiol levels are relatively low, and progesterone levels are high. The critical time, therefore, that the exogenous protease inhibitors are needed is on pregnancy day 0.5 when there are the most differences between the WT and cKO oviductal environments. A statement to this effect has been added to the Discussion (end of ninth paragraph).

We also agree that the graph representing the embryo development data in Figure 7 could be somewhat difficult to follow, and as a result we have changed the graph format. The stacked bars in the original graph represented in white the morula/blastocyst embryos, and in black the underdeveloped embryos. So, in the cKO-PI group, there were ~30% underdeveloped and 70% morula/blast, while in the cKO + PI group there were ~10% underdeveloped and 90% morula/blast. All mice in both WT groups had 100% morula/blast, and no underdeveloped embryos. We have now un-stacked the bars to make the graph clearer. The numbers of embryos per group were moved into the Figure legend.

*11) The conclusions about effects on the innate immune system seem to somewhat of a stretch and should be modified.*

Please see response to question #9.